# Cranberry Research Progress: A Systematic Review of Chemical Composition, Pharmacological Mechanisms, Clinical Applications, and Nutritional Significance

**DOI:** 10.3390/ijms26199707

**Published:** 2025-10-06

**Authors:** Jiao Xu, Chenliang Li, Wei Wu, Lingyang Kong, Lijin Xiao, Wei Ma, Lihong Zhang

**Affiliations:** 1Jiamusi College, Heilongjiang University of Chinese Medicine, Jiamusi 154007, China; xujiao2023@163.com; 2Key Laboratory of Northern Medicine Basis and Application, Ministry of Education, Heilongjiang University of Traditional Chinese Medicine, Harbin 150040, China; 3School of Pharmacy, Heilongjiang University of Chinese Medicine, Harbin 150040, China; l7174080625@163.com (C.L.); wuwei52414@163.com (W.W.); hljkly970219@163.com (L.K.); 15260756338@163.com (L.X.)

**Keywords:** cranberry, phytochemistry, pharmacologic effects, clinical applications, nutrition

## Abstract

Cranberry (*Vaccinium macrocarpon* Ait.) is an herbaceous, evergreen, dwarf shrub of the genus *Vaccinium* in the family Ericaceae, often used as a functional food. Cranberries are primarily distributed in the northern United States—including Massachusetts, Wisconsin, and Maine—as well as in Quebec, Canada; the state of Columbia; Chile in South America; and northeastern Europe. They are also found in China’s Greater Khingan Range and Fuyuan City, Jiamusi, Heilongjiang Province. The plants thrive in cool environments and exhibit considerable adaptability to soil conditions, preferring acidic soils. Cranberries are rich in a variety of biologically active components, such as polyphenols (proanthocyanidins, chlorogenic acid, flavonols, anthocyanins, caffeic acid, etc.), triterpenoids, and other nutrients. Studies have shown that the chemical components extracted from cranberry fruit have pharmacological effects such as antioxidant, anti-inflammatory, anti-cancer, and urinary tract infection prevention and treatment, and are commonly used clinically in the treatment of cardiovascular diseases, the prevention of urinary tract infections, blood pressure lowering, and the fight against Helicobacter pylori, among other clinical diseases. Cranberries also play a huge role in daily nutrition, and they are named for their richness in a variety of mineral elements, trace elements and vitamins. This work uses information from Pubmed, Web of Science, Scopus, CNKI (China National Knowledge Infrastructure), and related papers. In this paper, a comprehensive review of the phytochemical composition, pharmacological mechanism of action, clinical application value and nutritional significance of cranberry was conducted in recent years to provide references for the further extraction of chemical components in cranberry and rational clinical application, which can help to guide people to rationalize their diets and promote the formation of healthy diets.

## 1. Introduction

Cranberry (*Vaccinium macrocarpon* Ait.) belongs to the genus *Vaccinium* within the family Ericaceae. It is an evergreen shrub characterized by rounded or obovate leaves, white or reddish campanulate flowers, and purplish-red berries [1]. Cranberries grow mainly in the cool, acidic soil of the northern hemisphere and are widely planted in natural ecological bases mainly in North America and Europe, as well as in environmental planting bases in China’s Daxing’anling region and Fuyuan City in China’s Heilongjiang Province. Cranberries are known as the three traditional North American fruits, along with Concord grapes and blueberries [2]. The Dictionary of Traditional Chinese Medicine records that cranberry leaves are diuretic and detoxifying and are commonly used for rheumatism and gout; the fruit can be used to treat dysentery. The American Pharmacopoeia records that cranberry has been used as an adjunct against cystitis and urinary tract infections, and the therapeutic effect has been widely recognized [3,4]. The results of chemical experimental analyses indicate that cranberries contain mainly proanthocyanidins, phenolic acids, flavonoids, vitamins, triterpenoids, trace and bulk elements, and other chemical constituents [5,6,7,8,9]. With the advancement of analytical techniques, researchers have more accurately extracted and analyzed the bioactive components and chemical constituents of cranberries, such as using high-performance liquid chromatography (HPLC) and liquid chromatography–mass spectrometry (LC-MS) to quantitatively analyze the triterpene constituents and polyphenolic chemical constituents of cranberries [10].

Modern studies in the field of pharmacology have revealed a variety of pharmacological activities of cranberry, including antioxidant, anti-inflammatory, antibacterial, antitumor, and cardiovascular disease prevention [11,12]. After years of in-depth research by scholars at home and abroad, it was found that the acidic components in cranberries were able to elevate the acidity of bladder urine, which in turn inhibited the growth of Escherichia coli through the creation of an acidic environment, which is considered to be a significant effect of cranberries in the prevention of urinary tract infections [13]. The diverse pharmacological effects of cranberries are closely related to their rich content of bioactive substances and chemical components, which include phenolic acids, proanthocyanidins, and anthocyanins. In addition, vitamin C, vitamin E, potassium, calcium and other nutrients contained in cranberries are also important for the maintenance of human health [14]. In terms of clinical applications, cranberries have a wide range of applications, mainly for the treatment of urinary tract infections, and have shown significant therapeutic potential in the areas of cardiovascular health, oral health, and gastrointestinal health [15]. Cranberries are available in a variety of preparations, including cranberry juice, cranberry powder, cranberry tablets, cranberry oral solution, cranberry mouthwash, and so on, and the appropriate form of preparation should be selected according to the patient’s specific situation during clinical treatment.

This study summarizes the progress of cranberry in terms of its bioactive and chemical components, pharmacological mechanism of action, clinical application value and nutritional applications, and discusses the current problems in cranberry research. The aim of this study is to provide a theoretical basis for the extraction of cranberry chemical components and their further development and utilization, to make up for the deficiency of cranberry extraction and separation technology, and to provide a reference for the development of cranberry preparations in the future.

## 2. Materials and Methods

### 2.1. Identification and Selection of Studies

The initial phase of our analysis involved systematically assessing all studies identified through keyword searches pertaining to Cranberry. Following the removal of duplicate entries, we conducted a preliminary review of titles and abstracts to evaluate their relevance based on the established inclusion criteria. For studies that satisfied these criteria, a detailed examination was performed, encompassing a thorough analysis of the full text and an in-depth review of the reference lists to ensure a comprehensive understanding of the relevant literature.

### 2.2. Search Strategy

We identified the studies independently using the following keywords: Cranberry. In addition, reported pharmacological activities and phytochemical compositions were searched as keywords. This study only includes results found before March 2025 (without time restrictions before this date). The search was carried out in the electronic bibliographic databases PubMed (https://pubmed.ncbi.nlm.nih.gov/, 15 March 2025), CNKI (http://www.cnki.net, 21 March 2025), Baidu Scholar (https://xueshu.baidu.com/, 23 March 2025), GoogleScholar (http://scholar-xm.top, 25 March 2025) and Traditional Chinese Medicine Baodian Network (www.ZYBD.com, 2 April 2025).

### 2.3. Inclusion and Exclusion Criteria

Our inclusion criteria encompassed all experimental studies investigating various aspects of Cranberry, including its pharmacological effects, clinical applications, and nutrition, irrespective of the extraction methods employed. Additionally, we incorporated Chinese dissertations and theses that detailed the properties of Cranberry. Studies performed in humans, editorials, conference abstracts, the contents of dissertations that have been published in scientific journals, and conference proceedings were excluded.

### 2.4. Illustration Methods

Data in the text references relevant pharmacological monographs and other sources (USDA FoodData Central, EFSA, CFIA). This paper reviews several aspects of cranberry chemical composition, pharmacological effects, clinical applications, and nutritional studies. Pharmacological action diagrams were drawn by Photoshop 2025, Adobe Illustrator 2023, and Power Point 2021; chemical structure formulas were drawn by KingDraw 3.0.

## 3. Results

### 3.1. Search Results

The preliminary search yielded a total of 11,253 records, distributed across various databases as follows: 2533 from PubMed, 144 from CNKI, and 8576 from Baidu Scholar. After removing 8622 duplicate entries, we conducted a thorough review of titles and abstracts, which led to the exclusion of an additional 184 reports. This left 2015 studies for further evaluation. Upon detailed examination of the full texts, 767 studies were excluded: 432 for involving cranberry in combination with other substances, 333 for being published prior to 2003, and 2 due to unavailability of the full text. Through this rigorous screening process, a final set of 106 articles was selected for inclusion (Figure 1).

### 3.2. Cranberry Chemical Composition

Cranberry fruit is rich in a variety of unique chemical constituents, including flavonoids, anthocyanins, phenolic acids and triterpenoids [16]. As an important source of bioactive components, cranberry fruit is also rich in dietary fiber, minerals (calcium, potassium, and magnesium), vitamin C, and vitamin E [17]. Domestic and foreign researchers have used a variety of technical means to extract the chemical composition of cranberries and found that they also contain soluble substances such as dietary fiber and pectin [18]. Cranberries are particularly rich in polyphenolic compounds, covering flavonols [19], proanthocyanidins [20], tannins [21], and phenolic acids and anthocyanins. Polyphenols are a class of natural organic compounds with multiple phenolic hydroxyl groups, which belong to the secondary metabolites of plants and play a key role in physiological processes such as plant growth and development and defense against pathogens [22,23]. Currently, the research results on the chemical composition of cranberry are scattered, and researchers have mainly utilized experimental techniques such as spectrophotometry, liquid chromatography, and high-performance liquid chromatography to conduct in-depth studies on the chemical composition of cranberry fruits [24,25]. The article summarizes extraction methods for different categories of chemical constituents in cranberries (Figure 2).

#### 3.2.1. Proanthocyanidins

Proanthocyanidins are a class of polyphenolic compounds that are extensively distributed in the plant kingdom. These compounds can be transformed into anthocyanins through heating in an acidic milieu. Proanthocyanidins in cranberries are mainly composed of A-type bonds, whereas proanthocyanidins from other plant sources are dominated by B-type bonds [26]. Proanthocyanidins, which are characteristic of cranberries, have a unique chemical structure and are oligomers or polymers formed by different amounts of catechins and epicatechins linked by C4-C8 bonds or C4-C6 bonds. Owing to its abundance of type A proanthocyanidins, cranberry can effectively impede the adhesion of Escherichia coli to uroepithelial cells [27]. Type A proanthocyanidins, serving as the characteristic components of cranberry that differentiate it from other berries, have manifested their distinctive anti-urinary tract infection activity [28]. Kühn et al. [29] used a ternary mixed solvent (CO_2_ + ethanol + water) at 25–50 MPa and the stepwise extraction of polyphenols and oils was achieved under 25–50 MPa and 40–60 °C. Compared with the traditional solvent method, the ethanol consumption was reduced by 88%, and high-purity proanthocyanidins were extracted from cranberry pomace. The polyphenolic chemical components in cranberry can be effectively separated by high performance liquid chromatography (HPLC) with appropriate columns and mobile phases. Özlem et al. [30] used microwave, ultrasound, and combined ultrasound–microwave techniques for the extraction of phenolic compounds from cranberries, and the results of the experiments included total monomeric anthocyanin content, polymer color, vitamin C content, and the extracts’ other concentrations of phenolic compounds. The proanthocyanidin component of cranberry is one of its core active ingredients with multi-targeted benefits, and its use requires strict dosage control. The proanthocyanidin components in cranberries were summarized and categorized, and the results are presented in Figure 3.

#### 3.2.2. Anthocyanin

The anthocyanin compounds abundant in cranberries not only impart rich color to the fruit and its processed foods but also hold potential health benefits for humans. Anthocyanins are water-soluble natural pigments widely distributed in the plant kingdom, and their hydrolysis products are colored glycosides. As important components of plant secondary metabolites, anthocyanins play a key role in the food industry and human nutrition. Six major anthocyanins have been identified in cranberries, including geranium pigment, cornflower pigment, delphinium pigment, peony pigment, petunia pigment, and mallow pigment [31]. It was noted that the anthocyanin content of cranberry rind was significantly higher than that of the fruit pulp, and the peak anthocyanin content of ripe cranberries, where the major anthocyanin components were anthocyanin-3-O-galactoside and anthocyanin-3-O-arabinopyranoside [32,33]. Abeywickrama et al. [34] found that anthocyanins were mainly present in paeonidin-3-O-arabinoside and paeonidin-3-O-galactoside in Canadian wild cranberries. The composition of anthocyanins in cranberries from different regions varied, and the main components of anthocyanins in cranberries from European regions were paeoniflorin-3-O-galactoside, paeoniflorin-3-O-arabinopyranoside, cornflowerin-3-O-galactoside, and cornflowerin-3-O-arabinopyranoside [35,36]. These findings suggest that the content of anthocyanin active components in cranberries is influenced by fruit ripening stage, origin and varietal differences. In addition, several in vitro and in vivo studies have revealed the potential benefits of anthocyanins and their anthocyanin-rich foods in the prevention of diabetes, cancer, and cardiovascular and neurodegenerative diseases, as well as the widespread interest in the fight against oxidative stress. The anthocyanin components in cranberries were summarized and categorized, and the results are presented in Figure 4. Additionally, we summarize the anthocyanin content in different cranberry varieties from References [16,18] in Table 1.

#### 3.2.3. Flavonols

Flavonols, a class of secondary metabolites widely distributed in the plant kingdom, play a key role in plant growth, development, and defense responses [37]. The flavonol components contained in cranberries mainly include quercetin, kaempferol, populin, isorhamnetin, myricetin, myricetin 3-O-galactoside, hyperpside, isoquercitrin, and quercetin 3-O-α-L-arabinoside, which exhibit significant physiological activities and biological values. Researchers have found that European cranberry pomace extracts are rich in stilbenes as well as more organic acids and flavonols than other fruits, with the content of populin determined to be 8.4 mg/100 g [38]. Further, by analytical techniques such as LC-MS/MS, researchers detected isorhamnetin in cranberries at 2.1 mg/g [39]. Isorhamnetin predominantly exists in cranberries in the form of glycosides, such as isorhamnetin-3-O-glucoside. In certain studies, acetylated and caffeoylated glycoside derivatives of isorhamnetin have been isolated. These derivatives may exert an influence on the physiological activity of cranberries. The caffeoyl groups enhance the electron-transfer capacity of molecules, endowing them with a greater efficacy in scavenging free radicals like DPPH and ABTS. Moreover, in comparison to unacylated isorhamnetin glycosides, these derivatives display more excellent inhibitory effects on lipid peroxidation. Such derivatives can more effectively safeguard cellular DNA and proteins from damage induced by oxidative stress. Especially in experiments simulating intestinal or vascular environments, they manifest more prominent inhibitory effects on oxidative-stress-related apoptosis. [40,41]. Flavonol components of cranberry often act synergistically with other polyphenolic components to enhance the clinical effect of inhibiting urinary tract infections. Different geographical environments and harvesting times have significant effects on the content of flavonols in cranberries [42]. Among the flavonols extracted from cranberries, quercetin was found to significantly inhibit the NF-κB pathway, which in turn reduced the inflammatory response and viral replication [43]. In addition, populins extracted from cranberries have been shown to modulate cell cycle blockade, induce apoptosis, inhibit angiogenesis and metastasis, and other pathways to exert anticancer effects, and show therapeutic impact on a variety of cancers, including lung, colon, and breast cancers [44,45,46,47]. Comprehensive multi-dimensional studies have analyzed the potential research value of flavonoid components in cranberries in the prevention of urinary tract infections and anti-inflammation. The flavonol components in cranberries were summarized and categorized, and the results are presented in Figure 5. Additionally, we summarize the flavonol content in different cranberry varieties from References [16,18] in Table 2.

#### 3.2.4. Phenolic Acid

Phenolic acids, a group of organic acids containing a phenolic ring structure, are widely distributed in a variety of plant foods such as fruits, vegetables, tea and coffee. In particular, fruits such as cranberries, blueberries and grapes are rich in a variety of phenolic acids. In cranberries, the major phenolic acids include chlorogenic acids [48], hydroxycinnamic acids [49], and hydroxybenzoic acids [50]. Among them, hydroxybenzoates have the highest content, especially benzoic acid, while 2,4-dihydroxybenzoic acid, p-hydroxybenzoic acid, and o-hydroxybenzoic acid have relatively low content [51]. The hydroxycinnamic acids in cranberries are mainly composed of p-coumaric acid, caprylic acid, caffeic acid and ferulic acid [52]. Researchers successfully isolated and characterized phenolic acids with significant differences in content in cranberries by gas chromatography–mass spectrometry (GC-MS) [53], which included benzoic acid, o-hydroxybenzoic acid, cinnamic acid, m-hydroxybenzoic acid, p-hydroxybenzoic acid, p-hydroxybenzeneacetic acid, o-phenylenedicarboxylic acid, 2,3-dihydroxybenzoic acid, vanillic acid, o-hydroxycinnamic acid, 2,4-dihydroxybenzoic acid, p-coumaric acid, ferulic acid, caffeic acid, and erucic acid. The content of phenolic acids in cranberries varied significantly between growing environments and varieties and, as important bioactive constituents, they exhibited a wide range of antioxidants, anti-inflammatory, and antimicrobial benefits (Table 3).

#### 3.2.5. Triterpenoids

Triterpenoids, as diverse components of plant metabolites, exhibit diverse physiological activities and have significant economic value [54]. The triterpenoid chemical components in cranberry, as key components of its plant secondary metabolites, also exhibit diverse physiological activities. The triterpenoids mainly contained in cranberry are categorized into two main groups: pentacyclic triterpenoids and tetracyclic triterpenoids. Wu et al. extracted and isolated the chemical components of triterpenoids in cranberry, and ursolic acid, oleanolic acid, β-sitosterol, and stigmasterol were successfully identified in the extracts [55]. Ursolic acid, the most intensively studied chemical constituent in cranberry, is distributed in the fruit, leaves, and seeds, and is especially high in the mature fruit [56]. Ursula acid was the dominant triterpenoid among all cranberry species with a range of 2.72–16.80 mg/g DW, followed by oleanolic acid with a range of 1.07–7.89 mg/g DW [57]. In addition, the pentacyclic triterpenoids extracted from cranberry include oleanolic acid, glycyrrhetinic acid, betulinic acid, and lupinol, which have different effects on the physiological activities of cranberry. The oleanolic acid chemical constituents in cranberries are mainly found in the fruit and epidermis, and their structure connects carboxyl and hydroxyl groups at the C-3 position, and often forms esters or glycosides at the C-28 position. Studies have indicated that oleanolic acid can improve high-fat diet-induced obesity by regulating the intestinal flora, while synergizing with cranberry polyphenols to enhance the bacteriostatic effect on Escherichia coli [58]. The tetracyclic triterpenoids extracted from cranberries are mainly dammarane-type and lanosterane-type, but in view of the low number of related research reports and only sporadic experimental studies proving their existence, systematic studies are needed in the future to confirm the existence of their chemical constituents and to provide scientific evidence for the field of medicine and food functionality. The triterpenoid components in cranberries were summarized and categorized, and the results are presented in Figure 6.

#### 3.2.6. Other Compounds in Cranberry

Cranberry, a berry with a long history of medicinal and culinary use, has received widespread attention for its unique nutritional composition and health benefits. The berry is rich in carbohydrates, mainly fructose and glucose, as well as dietary fiber, soluble pectin, and insoluble cellulose, which play key roles in regulating intestinal function and improving metabolism [59]. The major minerals in cranberries include manganese, potassium, and magnesium, as well as trace elements such as calcium, iron, and zinc. Manganese, in particular, is present in cranberries at low levels (0.05–0.1 mg/100 g), but as an essential trace element, it plays an indispensable role in important physiological processes such as bone formation, carbohydrate metabolism, and oxidative stress defense [60]. Studies have pointed out that the manganese content in wild cranberries is 50% higher than that in common varieties, and this phenomenon is closely related to the soil environment, with different soil conditions significantly affecting the content of trace elements and minerals in cranberries [61]. Given the diversity of mineral content in cranberry and its potential ability to act through multiple synergies, this berry can be considered as a natural resource for maintaining the metabolism of the organism and meeting daily nutritional requirements.

### 3.3. Cranberry Pharmacology

Cranberry, an herb belonging to the genus *Vaccinium* in the family Ericaceae, is rich in anthocyanins, proanthocyanidins, flavonols, and other bioactive substances. These components have shown remarkable efficacy in pharmacological effects such as anti-inflammatory, antioxidant, antibacterial, and antitumor effects, and have demonstrated potential applications in research areas such as metabolic regulation and skin health [62,63,64]. To date, a large number of studies have focused on the analysis of the chemical composition of cranberry, thus validating its pharmacological effects [65]. However, the current exploration of the pharmacological effects of cranberry is still shallow, and further in-depth studies on its pharmacological effects and its mechanism of action are urgently needed. The article provides a summary of the principal chemical constituents of cranberries and their corresponding pharmacological effects, as depicted in Figure 7.

#### 3.3.1. Antioxidant Activity

Cranberry fruits, as a major source of antioxidants, are rich in chemical components such as proanthocyanidins, anthocyanins, phenolic acids, ascorbic acid and triterpenoids, which exhibit significant antioxidant activity in cranberries. These chemical components exert their powerful antioxidant effects by scavenging excess free radicals in the body, such as hydroxyl radicals and superoxide anion radicals [66]. Free radicals, as unstable molecules produced during the body’s metabolism, and their excessive presence can attack intracellular lipids, proteins, and DNA, leading to cellular senescence and apoptosis [67]. When the body is in a pathological state such as inflammation, ischemia–reperfusion, or hyperlipidemia, NO production becomes abnormal. Flavonoids (anthocyanins, etc.) in cranberries may positively affect human health by inhibiting oxidative stress, modulating NO levels, and exhibiting significant cardioprotective and neuroprotective effects [68]. Studies have revealed that components such as delphinidin and cornflower pigments extracted from cranberries have antioxidant properties that reduce intracellular ROS levels and relax ciliary muscles [69]. Further studies have shown that cranberry extracts can significantly increase the activity of antioxidant enzymes in the body, strengthen the body’s antioxidant defense system, reduce the damage caused by oxidative stress on the body, and prevent and delay the occurrence of diseases [70,71]. Samir C et al. found that there were significant differences in the content of biologically active components in different varieties of wild cranberries by using DNA probe labeling technology, and that there were significant differences in the antioxidant capacity of different cranberry varieties [72]. The antioxidant activity of cranberries is attributed to their diverse polyphenolic composition and their unique bioactive mechanism, which makes cranberries have a wide range of applications in the field of food, nutraceuticals and drug development (Figure 8).

#### 3.3.2. Antimicrobial Activity

Cranberry exhibits significant antimicrobial activity, especially in the inhibition of common pathogenic bacteria in the urinary system, and its unique antimicrobial mechanism is valuable in the prevention and adjuvant treatment of related diseases. Cranberries are rich in proanthocyanidins, anthocyanidins, triterpenoids, and other active components, which exert their antimicrobial effects by decreasing bacterial adhesion and disrupting the cell wall or cell structure [73]. Streptococcus pyogenes, as the main cariogenic bacteria in the oral cavity, significantly affects oral health. Bioactive components such as type A proanthocyanidins in cranberry extracts have inhibitory effects on the growth of Streptococcus pyogenes, inhibit the formation of insoluble polysaccharides in the extracellular matrix, and prevent glycan-mediated adhesion, cohesion, and aggregation of proteins in Streptococcus pyogenes [74]. Furthermore, cranberries inhibit the adhesion of Porphyromonas gingivalis and Fusobacterium nucleatum to gingival epithelial cells, thereby reducing the inflammatory response caused by bacterial invasion of gingival tissue. In vitro experimental studies have shown that a variety of chemical components in cranberry extracts have significant inhibitory effects on a wide range of pathogenic bacteria, suggesting that they could be an important source of antimicrobial foods [75]. It was found that in a urinary barrier model of T24 cells grown in Transwell inserts and either uninfected or UPEC-infected, treatment with cranberry polyphenol extracts 3,4-dihydroxyphenylacetic acid, and phenylacetic acid showed favorable changes in barrier permeability and integrity [76]. Cranberry has been widely used in the prevention of urinary tract infections and contains components such as proanthocyanidins that can alter the structure of bacterial trichomes and prevent them from binding to receptors in the urethral epithelium, thus reducing the proliferation of bacteria in the urethral epithelium and decreasing the risk of infection [77]. Proanthocyanidins extracted from cranberries can interfere with the adhesion of Helicobacter pylori on the surface of gastric mucosa, inhibit its growth and reproduction, and play a protective role in gastric health [78]. The antimicrobial activity of cranberry is mainly attributed to the mechanism of action of the polyphenolic constituents it contains. It has a clear potential for application in the prevention of urinary tract infections and oral health in particular. However, the optimization of the extraction process of its constituents and further studies on clinical treatment protocols still need to be explored in depth (Figure 9).

#### 3.3.3. Anti-Inflammatory Activity

Cranberries are rich in bioactive components that allow them to exhibit significant anti-inflammatory properties, and their anti-inflammatory effects are attributed to the modulation of inflammatory cell signaling pathways. Activation of the pathway of NF-κB, a key regulator during inflammation, promotes the release of inflammatory factors [79]. Polyphenolic compounds in cranberries exhibit significant anti-inflammatory effects, which may be related to the inhibition of reactive oxygen species (ROS) production and interference with cellular signaling pathways, resulting in anti-inflammatory effects [80]. Terpenoids in cranberry have been shown to significantly inhibit the expression of pro-inflammatory cytokines (IL-6, tumor necrosis factor TNF-α) in THP-1 cells, a human monocyte cell line, inhibiting cellular pathways for anti-inflammatory effects [81]. Several studies have evaluated the anti-inflammatory effects of cranberry and its ability to enhance immunomodulation in periodontitis and dental implant therapy, and the results suggest that its anti-inflammatory effects may be related to the clearance of inflammatory factors by macrophages [82,83,84,85]. In addition, cranberry juice was able to reverse rotenone (ROT)-induced oxidative stress in multiple organs, especially in the brain where the inhibitory effect was most pronounced [86]. Thus, the anti-inflammatory activity of cranberry is a result of the direct action of its polyphenolic chemistry with the indirect control of gut flora metabolism and involves the inhibition of oxidative stress, the intervention of specific signaling pathways, and shows great potential in models of inflammation such as oral, gastrointestinal, and neurological (Figure 10).

#### 3.3.4. Anti-Tumor Activity

Cranberries are rich in a variety of bioactive components, such as proanthocyanidins, anthocyanidins, and triterpenoids, which confer potential antitumor activity. As a complex disease characterized by abnormal cell proliferation and metastasis, cancer affects multiple organs and systems throughout the body, and the number of cancer patients continues to rise globally, making the discovery of novel anticancer drugs particularly urgent [83]. It was noted that after treatment of human prostate cancer cells with different concentrations of cranberry extract for 6 h, cancer cell viability was significantly decreased, and the proportion of G2-M phase of the cell cycle was decreased and the proportion of G1 phase was increased, which may be related to the effect of cranberry extract on cell cycle regulatory proteins [54]. In addition, after treatment of lung cancer cells with cranberry extract, the proliferation of lung cancer cells was assessed by cell clonogenic morphology assay and cell proliferation assay, and the results showed a significant reduction in the number of lung cancer cells, which was correlated with the inhibition of the expression of the pro-proliferative molecules, cytokinin E, cytokinin-dependent kinase 2, and cytokinin-dependent kinase 4 by the cranberry extract. Bai Q. et al. found that the use of alone cranberry extract had little effect on U87 cells in human glioblastoma cell lines, but supplementation of cranberry extract in radiotherapy treatment significantly inhibited the proliferation of U87 cells and promoted apoptosis [87]. These findings suggest that cranberry and its extracts have the potential to be developed as natural antitumor agents, especially in tumor prevention and adjuvant therapy [88]. However, most of the experimental studies are still at the stage of animal testing, and clinical experimental studies need to be further validated (Figure 11).

#### 3.3.5. Other Pharmacological Effects of Cranberry

In addition to these pharmacological effects, cranberries have demonstrated other potential pharmacological activities. Cranberry has also shown positive effects in the treatment of cardiovascular health and neurodegenerative diseases [89]. Injections of cranberry extract into a rat model of cardiac injury have been shown to significantly improve serum markers associated with cardiac injury, promote low-density lipoprotein (LDL) oxidation, and enhance high-density lipoprotein (HDL) function, thereby reducing the risk of atherosclerosis [90]. In addition, the anti-inflammatory and antioxidant properties of cranberries help to alleviate oxidative stress and inflammatory damage in the brain and have preventive and ameliorative effects on neurodegenerative diseases such as Alzheimer’s disease and Parkinson’s syndrome [91]. However, cranberries may still have undiscovered pharmacological effects, which need to be explored in more in-depth experimental studies to realize their greater potential in the field of healthcare.

### 3.4. Clinical Applications of Cranberries

Cranberry, a type of berry abundant in diverse bioactive constituents such as proanthocyanidins, anthocyanidins, and terpenoids, which endow cranberries with notable anti-inflammatory, antioxidant, and antimicrobial pharmacological attributes, has garnered substantial attention within the realm of clinical applications. Clinical utilization of cranberry has manifested its latent value in multiple domains, particularly in urinary health and oral health. Moreover, cranberries have demonstrated potential advantages in the prevention of cardiovascular diseases and cancer. Nevertheless, during clinical application, individual variability and the interactions of cranberry products with other medications and therapies must be taken into account to guarantee the safety and efficacy of cranberry products.

#### 3.4.1. Urinary Health

Cranberry has significant clinical applications in the field of urologic disease prevention and treatment, especially in the prevention of urinary tract infections and adjuvant treatment of urinary tract diseases. Cranberry is rich in A-type proanthocyanidins, which exhibit significant inhibitory effects on urinary tract pathogenic P-fimbriated E. coli and effectively reduce bacterial adherence to the cell epidermis [87]. Studies have shown that cranberry juice is effective in preventing urinary tract infections due to the presence of two compounds with anti-adhesion properties that prevent adhesion of fimbriae-expressing E. coli to urinary tract epithelial cells [88]. Cranberry intake also reduces symptoms associated with urinary tract infections, such as urinary frequency, urgency, purulent exudate, and abnormal odor by inhibiting the inflammatory response to urinary tract infections. In addition, cranberry helps to maintain the balance of the urethral flora, regulate the pH of the urethral environment, inhibit the growth of harmful bacteria, and promote the proliferation of beneficial bacteria, which further strengthens the natural defense mechanism of the urethra and prevents the occurrence of infections. Cranberry and its extracts can directly acidify urine, improve the urethral environment, and inhibit the reproduction of urethral pathogenic bacteria due to its lower pH value [89]. Proanthocyanidins and their microbial metabolite 3,4-dihydroxyphenylacetic acid from cranberries partially reversed the pro-inflammatory microRNA response in intestinal epithelial cells, which in turn affects the systemic circulation and consequently the urogenital microenvironment [90]. Microbial metabolites of proanthocyanidin-rich cranberry extract significantly attenuated the activation of E. coli virulence genes during the early intestinal phase, and intestinal virulence altered the microbiota-dependent manner in which the urinary epithelium is infected [91]. Therefore, cranberry maintains the health of the urinary tract through multiple mechanisms of action such as acidification of urine, inhibition of adhesion of pathogenic bacteria, and regulation of flora metabolism, but its effects are affected by the composition of the individual flora and the content of proanthocyanidins and other biologically active components, as well as by the different disease states, and in future further investigations should be undertaken in the modulation of the effects of cranberry on specific flora and in different populations.

#### 3.4.2. Oral Health

Oral health, as an important part of human health, is closely linked to general health and quality of life. Cranberry, as a fruit rich in a variety of bioactive substances, such as proanthocyanidins and anthocyanins, has a positive effect on oral health by preventing and treating dental caries, decreasing dentin erosion, and promoting tooth remineralization [92]. The chemical composition of proanthocyanidins contained in cranberries exhibits powerful pharmacological activities such as antioxidant and antibacterial, showing significant antioxidant effects on the growth of bacteria secondary to pulpal infections, as well as significant antimicrobial effects on bacterial infectious oral diseases such as dental caries and periodontitis [93]. It was noted that a caries incidence experiment was conducted on 60 preschool children, and it was found that daily intake of tablets containing probiotics and cranberry extract reduced the incidence of dental caries in preschool children [94]. In addition, cranberries and their juices, among others, can inhibit matrix metalloproteinases (MMPs) present in dentin and saliva, thereby reducing their erosion of dentin [95]. Cranberry extract enhances remineralization of enamel and dentin and exerts anticariogenic effects by decreasing the biomass and increasing the pH of dental biofilm [96]. In addition, cranberry positively affects oral health through various pathways, such as reducing plaque production and decreasing oral odor [97]. Cranberry has significant value in the prevention and treatment of oral diseases through antimicrobial, anti-biofilm, and regulation of microecological stabilization, etc. Current experimental studies have only preliminarily verified that cranberries have therapeutic effects on periodontitis, dental caries, and other oral infectious diseases, but it is still unclear whether cranberries have such therapeutic effects in clinical treatment. Therefore, more experiments are needed to prove the therapeutic effects of cranberries in clinical practice. However, its benefits on the oral aspect cannot replace conventional oral hygiene and health care measures, and the combination of moderate consumption of cranberries and good oral hygiene habits in daily life can help to better maintain oral health.

### 3.5. Nutritional Significance

Cranberries not only contain a variety of bioactive components, but are also rich in a variety of nutrients, including vitamins such as vitamin C, vitamin E, and B vitamins, as well as minerals such as potassium, magnesium, and calcium [98]. Through synergistic effects, these components can effectively remove free radicals from the body and protect cells from oxidative damage, which is important for the prevention of cardiovascular disease, cancer, and neurodegenerative diseases. In addition, the rich dietary fiber in cranberries helps to promote intestinal peristalsis and maintain the health of the digestive system [99]. The antioxidant properties of cranberries are mainly derived from their abundance of fat-soluble and water-soluble vitamins, of which ascorbic acid is the main form of vitamin C [100]. Cranberries are rich in polyphenols, flavonoids, terpenoids, and other compounds that confer significant antioxidant activity, which helps to prevent oxidative stress and a variety of diseases [101]. In terms of cardiovascular system prevention, cranberries are able to regulate cholesterol levels, prevent atherosclerosis, and reduce the risk of stroke, cerebral infarction, and heart disease [102,103]. In addition, cranberries have different roles in oral health prevention, slowing down aging, and enhancing the body’s immunity. Studies have shown that intake of cranberry beverage for 6 weeks significantly reduced UV-induced skin erythema and effectively improved the elasticity and smoothness of the skin on the face and arms, especially in women over 40 years of age [104]. The combination of cranberry extracts with other plant extracts can enhance the anti-aging effect through multi-component synergistic effects. The combination of cranberry and rosemary was found to slow down the oxidization process of refrigerated fresh meat, suggesting that cranberry complexes have a significant potential in the antioxidant process [105]. In conclusion, cranberries have a rich value in the field of nutrition, covering several aspects, and their rich nutritional value and benefits make them an extremely beneficial fruit for human health. Intake of cranberry fruits or cranberry compound products in daily life helps to maintain a healthy state. The data in the table were obtained from the composition of cranberry fruits and juices listed on the official website of the USDA Agricultural Research Service (https://fdc.nal.usda.gov/food-details/2003594/nutrients, 15 March 2025), and the content table is based on the content of *macrocarpon* as an example (*Vaccinium macrocarpon*) (Table 4).

## 4. Conclusions

Cranberry has achieved notable research results in phytochemistry, pharmacology, clinical medicine, and nutrition. In phytochemistry, cranberries contain various compounds like proanthocyanidins, anthocyanidins, triterpenoids, and nutrients such as vitamins and trace elements, which form the basis of their bioactivity. The high content of bioactive substances gives cranberries diverse pharmacological effects, including antibacterial, antioxidant, anti-tumor, and anti-inflammatory activities, and they are effective in preventing cardiovascular and urinary tract diseases. In clinical applications, cranberries show remarkable efficacy in preventing and treating urinary system diseases, maintaining gastrointestinal health, and intervening in cardiovascular diseases, providing new options for relevant disease prevention and treatment. In nutritional applications, cranberries are rich in nutrients, can be consumed in multiple ways, are suitable for all ages, and are widely used in nutritional diets.

However, contemporary research on cranberries has limitations. In the study of the mechanism of action, although the diverse effects of cranberries are well-known, the specific mechanism remains unclear. Some studies have small sample sizes and short research times, limiting the reliability and generalizability of experimental investigations. Different experimental methods and conditions make it difficult to compare and analyze results directly. Therefore, future research should focus on the mechanism of the pharmacological effects of cranberries to clarify how they treat diseases. At the same time, extensive and long-term clinical trials should be conducted to improve the reliability and credibility of experimental research. It is hypothesized that as research progresses, cranberries will assume a significant role in a broader range of fields and contribute more substantially to human health.

## Figures and Tables

**Figure 1 ijms-26-09707-f001:**
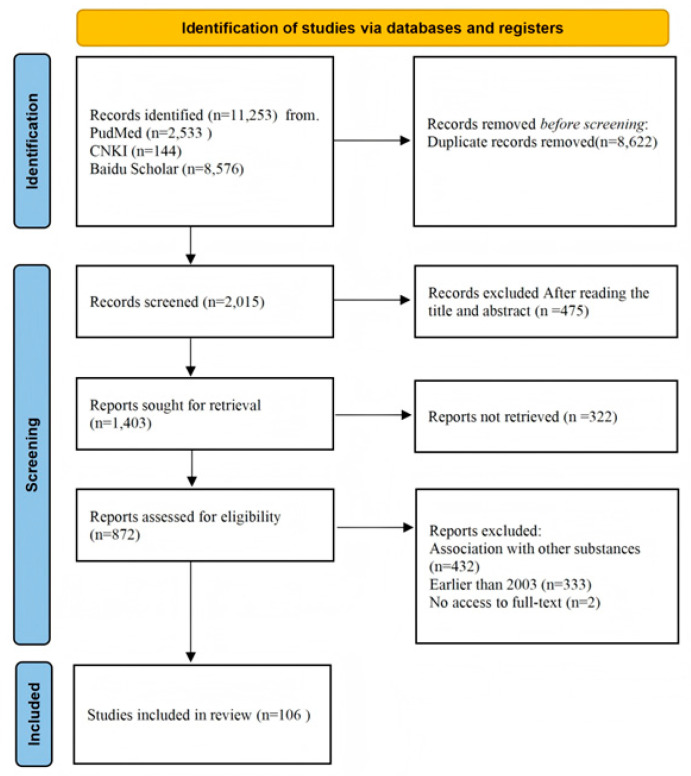
Flowchart of included studies in the review.

**Figure 2 ijms-26-09707-f002:**
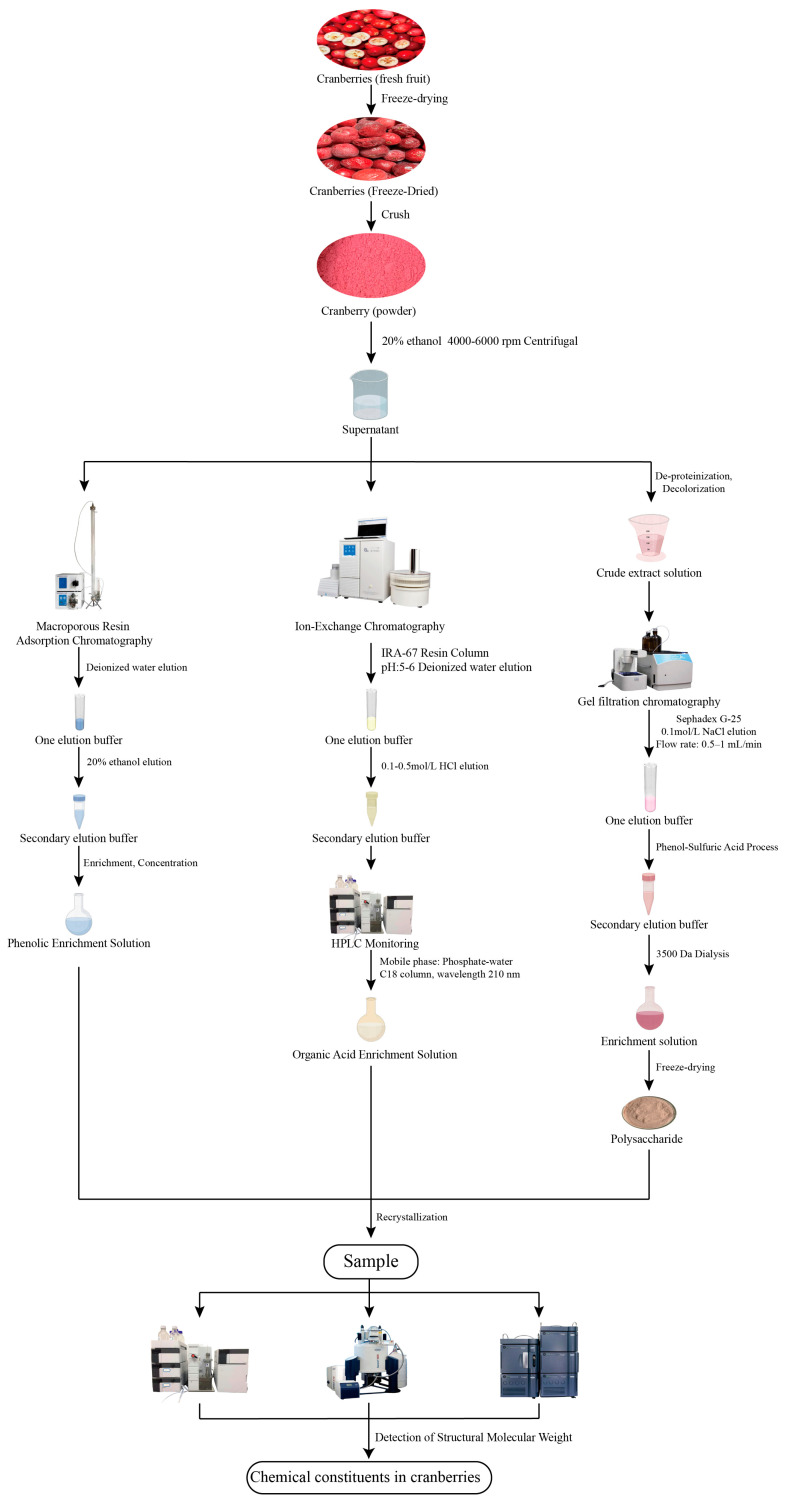
Extraction Process of Major Chemical Components in Cranberries.

**Figure 3 ijms-26-09707-f003:**
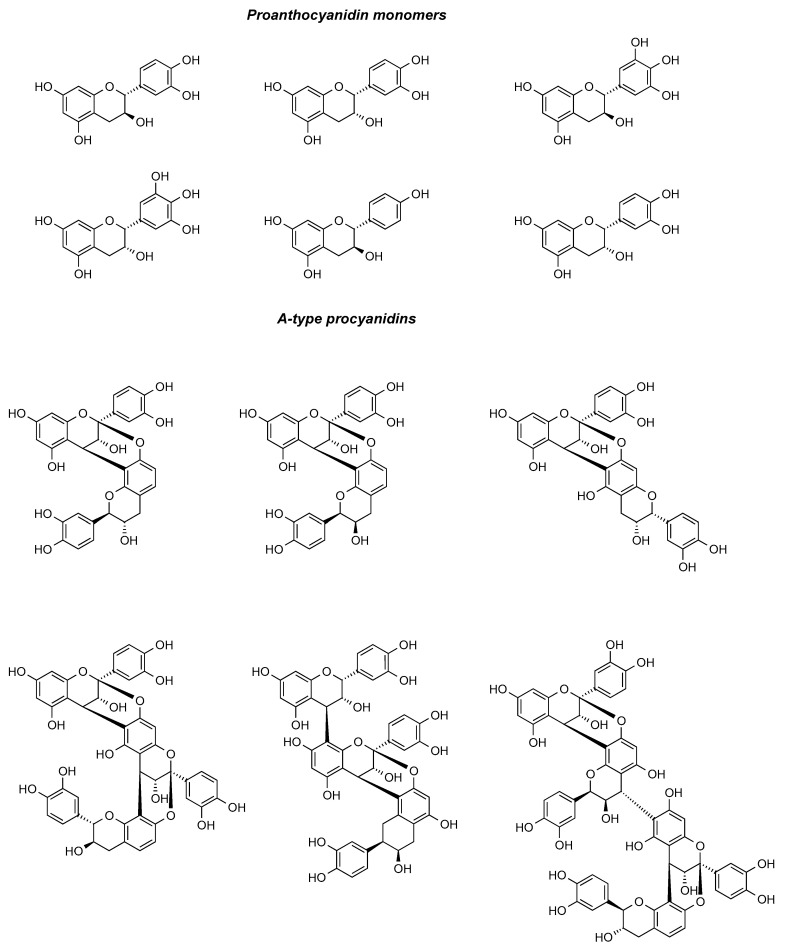
Major proanthocyanidin-like chemical constituents in cranberries.

**Figure 4 ijms-26-09707-f004:**
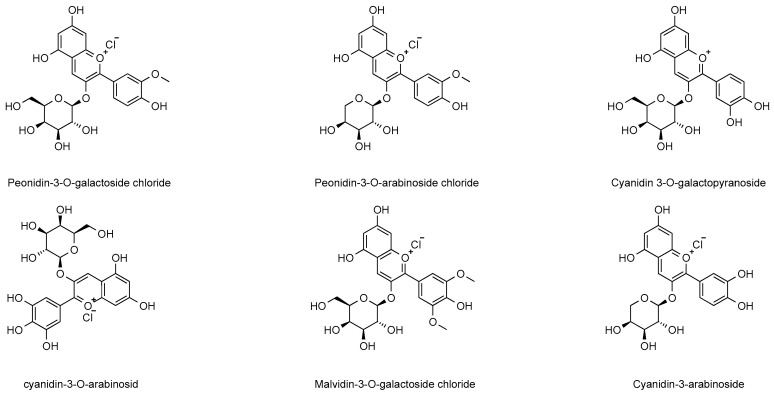
Major anthocyanin-like chemical constituents in cranberries.

**Figure 5 ijms-26-09707-f005:**
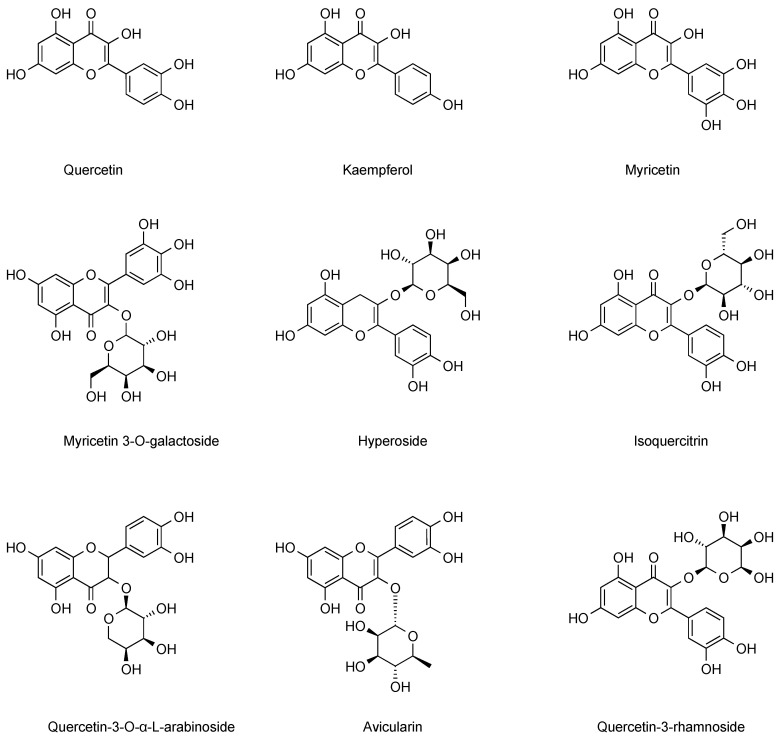
Major flavonol chemical constituents in cranberries.

**Figure 6 ijms-26-09707-f006:**
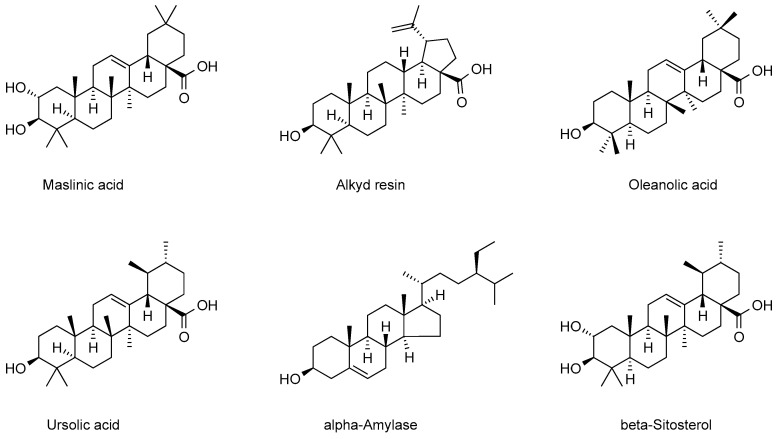
Major triterpenoid chemistry in cranberries.

**Figure 7 ijms-26-09707-f007:**
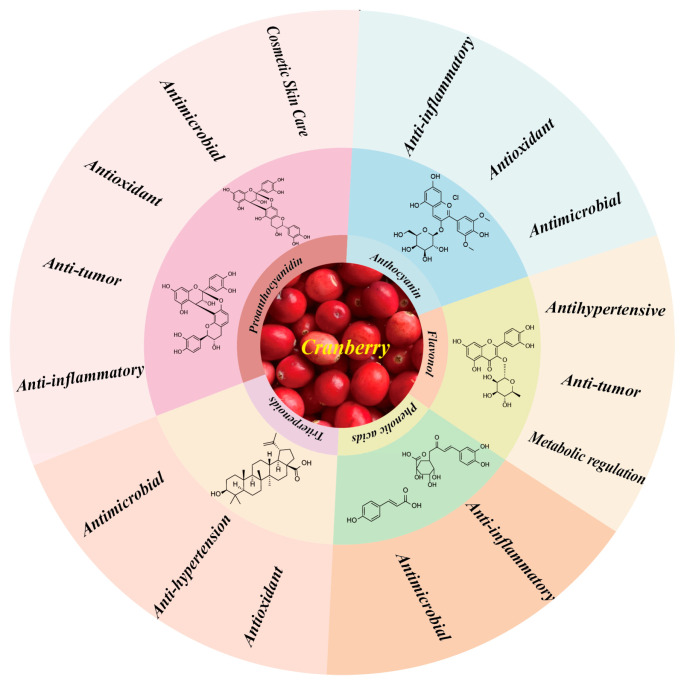
Main active ingredients and pharmacological effects in cranberries.

**Figure 8 ijms-26-09707-f008:**
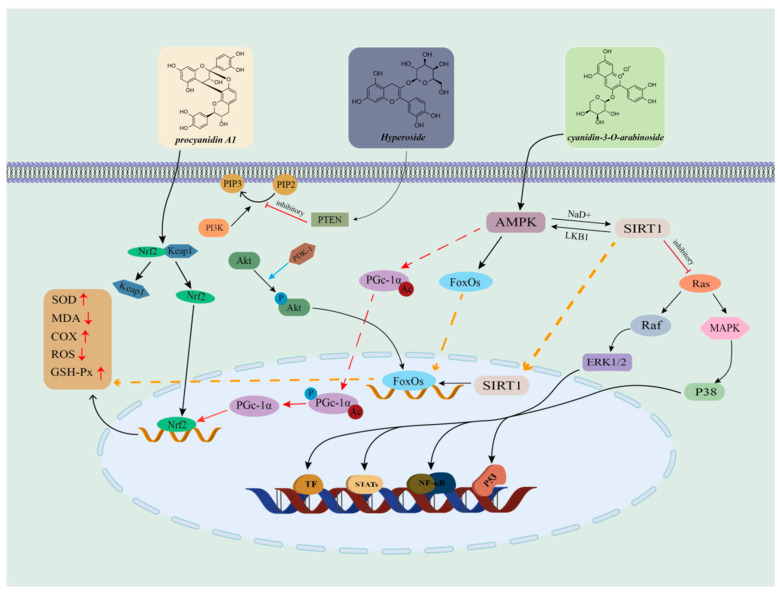
Mechanism of action of major antioxidant components in cranberry. (The three active components of cranberries work synergistically through multiple pathways, including ‘antioxidant (Nrf2) + metabolic regulation (AMPK/PI3K) + inflammation inhibition (MAPK),’ ultimately enhancing cellular antioxidant capacity, improving metabolism, and inhibiting inflammation.)

**Figure 9 ijms-26-09707-f009:**
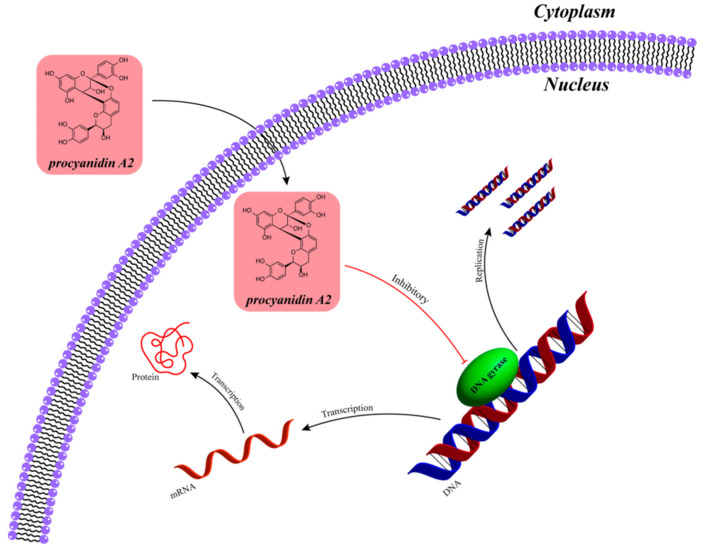
Schematic diagram of inhibition of bacterial protein synthesis by proanthocyanidin components in cranberries. (Cranberry proanthocyanidin A2 exerts its effects through the pathway of ‘inhibiting DNA gyrase→disrupting nucleic acid metabolism→inhibiting gene expression and cell proliferation.)

**Figure 10 ijms-26-09707-f010:**
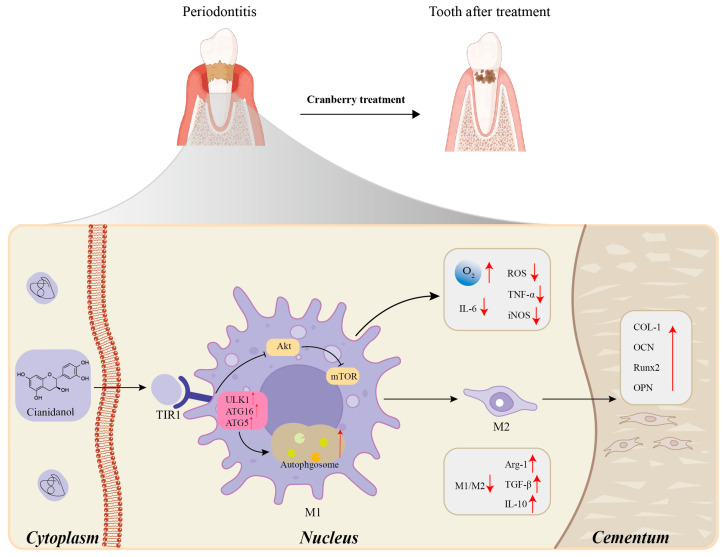
Mechanism of action of proanthocyanidin components of cranberries in the treatment of periodontitis. (Cyanidinol in cranberries precisely inhibits the ‘inflammation-destruction’ cycle of periodontitis through a cascade effect of ‘receptor binding→autophagy regulation→immune polarization→tissue regeneration,’ while simultaneously initiating the repair process.)

**Figure 11 ijms-26-09707-f011:**
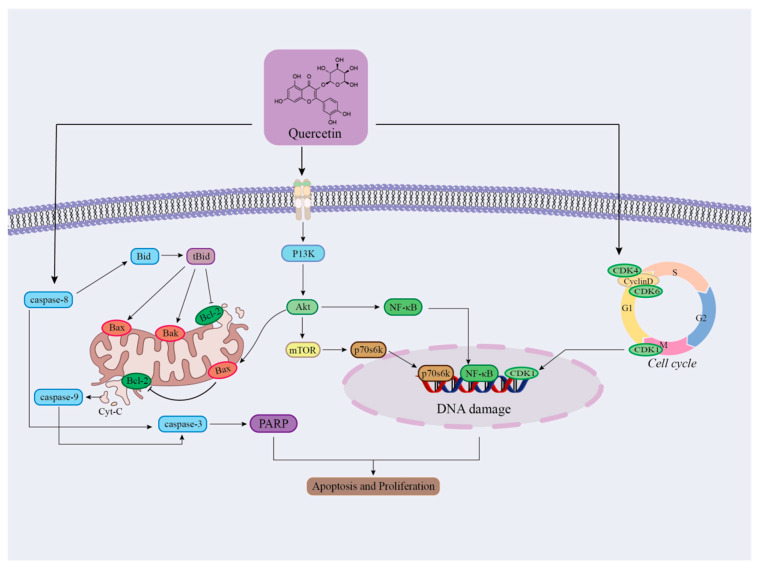
Schematic diagram of the mechanism of regulation of apoptosis and cell proliferation by flavonoid components in cranberry. (Quercetin forms a synergistic network through three dimensions: mitochondrial apoptosis, proliferation signal inhibition, and cycle arrest, ultimately achieving the effect of inhibiting cell survival and proliferation.)

**Table 1 ijms-26-09707-t001:** Major anthocyanin compounds in cranberries and their content across different varieties.

No.	Compound Name	Molecular	Content			References
			Pilgrim	Stevens	Ben Lear	
1	Delphinidin 3-O-glucoside	C_21_H_21_O_12_^+^	1.2 mg/100 g dm	0.6 mg/100 g dm	1.1 mg/100 g dm	[16]
2	Cyanidin 3-O-galactoside	C_21_H_21_O_11_^+^	116.5–118.9 mg/100 g dm	185–191.2 mg/100 g dm	168.8–172.4 mg/100 g dm	[18]
3	Cyanidin-3-O-glucoside	C_21_H_21_ClO_11_	5.3–5.5 mg/100 g dm	7.2 mg/100 g dm	7.9 mg/100 g dm	[16]
4	Cyanidin-3-O-arabinoside	C_20_H_19_ClO_10_	59–60.2 mg/100 g dm	94.6–97.8 mg/100 g dm	81.6–83.4 mg/100 g dm	[16]
5	Peonidin 3-galactoside	C_22_H_23_ClO_11_	190.1–192.7 mg/100 g dm	275.4–281.2 mg/100 g dm	374.4–381 mg/100 g dm	[16]
6	Peonidin 3-Glucoside	C_22_H_23_ClO_11_	20.6–21 mg/100 g dm	30.4–31.2 mg/100 g dm	50.1–50.7 mg/100 g dm	[18]
7	Peonidin 3-arabinoside	C_21_H_21_ClO_10_	42.3–42.9 mg/100 g dm	92.5–94.3 mg/100 g dm	113.3–115.7 mg/100 g dm	[16]
8	Malvidin 3-O-arabinoside	C_22_H_23_O_11_	1.2–1.4 mg/100 g dm	0.8 mg/100 g dm	1.7 mg/100 g dm	[16]

Note: dm: dry matter.

**Table 2 ijms-26-09707-t002:** Major flavonol compounds in cranberries and their content across different varieties.

No.	Compound Name	Molecular	Content			References
			Pilgrim	Stevens	Ben Lear	
1	Myricetin-3-O-galactoside	C_21_H_20_O_13_	343.6–351 mg/100 g dm	262.2-mg/100 g dm	146.6–150.4 mg/100 g dm	[16]
2	Myricetin-3-O-glucoside	C_21_H_20_O_13_	6.4–6.6 mg/100 g dm	3.8 mg/100 g dm	1.9 mg/100 g dm	[16]
3	Myricetin-3-O-xyloside	C_20_H_18_O_12_	28.6–29.4 mg/100 g dm	14.8–15.2 mg/100 g dm	6.3–6.5 mg/100 g dm	[16]
4	Quercetin-3-O-galactoside	C_21_H_20_O_12_	375.1–380.1 mg/100 g dm	312.5–318.7 mg/100 g dm	287.7–293.9 mg/100 g dm	[16]
5	Myricetin-3-O-glucuronide	C_21_H_18_O_14_	43.6–44.4 mg/100 g dm	27.4–28 mg/100 g dm	17–17.4 mg/100 g dm	[18]
6	quercetin-3-O-pentosyl-pentoside	C_25_H_26_O_15_	41–42.2 mg/100 g dm	38.9–39.7 mg/100 g dm	41.7–42.7 mg/100 g dm	[18]
7	quercetin-3-O-deoxyhexosyl(1-2)deoxyhexoside	C_27_H_30_O_15_	25.6–26 mg/100 g dm	20.4–20.6 mg/100 g dm	17.2–17.6 mg/100 g dm	[18]
8	3-O-Methylquercetin	C_16_H_12_O_7_	55.1–56.6 mg/100 g dm	57.1–58.7 mg/100 g dm	33.6–34.2 mg/100 g dm	[16]
9	Quercetin-3′-O-glucoside	C_21_H_20_O_12_	12.3–12.5 mg/100 g dm	8.5–8.7 mg/100 g dm	3.9–4.3 mg/100 g dm	[16]
10	Quercetin-3-(3-p-coumaroylglucoside)	C_30_H_26_O_14_	2.3 mg/100 g dm	1.6 mg/100 g dm	0.8–1 mg/100 g dm	[16]
11	Quercetin-3-O-rhamnoside	C_21_H_20_O_11_	8.9–9.1 mg/100 g dm	6.8–7.4 mg/100 g dm	5.9–6.1 mg/100 g dm	[16]

Note: dm: dry matter.

**Table 3 ijms-26-09707-t003:** Major phenolic acid chemical constituents in cranberries.

No.	Name	Content	Analytical Method	Reference
1	p-Coumaric acid	2–245 µg/g dw	HPLC/ESI-MS/MS	[16]
2	p-Coumaroyl hexose	8.6–13.9 mg/100 g dm	LC/MS Q-TOF and UPLC-PDA-FL	[16]
3	p-Coumaroyl hexose isomer	3.6–50.0 mg/100 g dm	LC/MS Q-TOF and UPLC-PDA-FL	[16]
4	p-Coumaroyl derivatives	210–451 mg/100 g dm	LC/MS Q-TOF and UPLC-PDA-FL	[16]
5	Chlorogenic acid	72.00–129.62 mg/100 g dm	LC/MS Q-TOF and UPLC-PDA-FL	[16]
6	Caffeic acid	5–123 µg/g dw	HPLC/ESI-MS/MS	[16]
7	Caffeoyl hexoside	92.7–190.2 mg/100 g dm	LC/MS Q-TOF and UPLC-PDA-FL	[16]
8	Caffeoyl hexoside isomer	10.9–17.5 mg/100 g dm	LC/MS Q-TOF and UPLC-PDA-FL	[16]
9	Caffeoyl and derivatives	39.93–68.28 mg/100 g dm	LC/MS Q-TOF and UPLC-PDA-FL	[16]
10	Ferulic acid	4–39 µg/g dw	HPLC/ESI-MS/MS	[16]
11	Total phenolic acid	327–649 mg/100 g dm	LC/MS Q-TOF and UPLC-PDA-FL	[16]

Note: dw: dry weight; dm: dry matter.

**Table 4 ijms-26-09707-t004:** Table of nutrient and trace element composition in cranberries.

Name	Average Amount	Number of Samples	Min	Max	Median
Proximates					
Water	92.3 g	8	92.1 g	92.5 g	92.3 g
Total lipid (fat)	0.34 g	8	0.16 g	0.86 g	0.28 g
Ash	0.1 g	8	0 g	0.15 g	0.1 g
Protein	0		0 g	0 g	0 g
Nitrogen	<0.02	8			
Carbohydrates					
Carbohydrate	7.26 g				
Sugars, Total	3.41 g				
Sucrose	<0.25 g				
Glucose	2.7 g	8	2.45 g	3.03 g	2.62 g
Fructose	0.72 g	8	0.61 g	0.89 g	0.68 g
Lactose	<0.25 g	8			
Maltose	<0.25 g	8			
Galactose	<0.1 g	8			
Organic acids					
Citric acid	1160 mg	8	997 mg	1520 mg	1100 mg
Malic acid	756 mg	8	615 mg	912 mg	764 mg
Quinic acid	1020 mg	8	821 mg	1220 mg	1020 mg
Oxalic acid	<40 mg	8			
Minerals					
Calcium, Ca	7 mg	8	5 mg	10 mg	5 mg
Iron, Fe	0.13 mg	8	0.09 mg	0.19 mg	0.11 mg
Magnesium, Mg	4.4 mg	8	3.6 mg	6.6 mg	3.9 mg
Phosphorus, P	3 mg	8	3 mg	4 mg	3 mg
Potassium, K	71 mg	8	61 mg	91 mg	81 mg
Sodium, Na	6 mg	8	2 mg	12 mg	5 mg
Zinc, Zn	0.04 mg	8	0.03 mg	0.05 mg	0.04 mg
Copper, Cu	0.013 mg	8	0 mg	0.022 mg	0.014 mg
Manganese, Mn	0.221 mg	8	0.119 mg	0.271 mg	0.228 mg
Vitamins and Other Components					
Vitamin C	<0.63 mg	8			
Thiamin	0.018 mg	8	0.015 mg	0.023 mg	0.017 mg
Niacin	0.01 mg	8	0.006 mg	0.013 mg	0.01 mg
Vitamin B-6	0.009 mg	8	0 mg	0.011 mg	0.01 mg
Folate, total	9 µg	8	0 µg	21 µg	8 µg

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
