# Peer review of "Cranberry Research Progress: A Systematic Review of Chemical Composition, Pharmacological Mechanisms, Clinical Applications, and Nutritional Significance"

_ijms, 2025, doi:10.3390/ijms26199707_

Round 1

Reviewer 1 Report

Comments and Suggestions for Authors

In this manuscript, Jiao Xu and colleagues presented a comprehensive review of the phytochemical composition, pharmacological mechanism of action, clinical application value and nutritional significance of cranberry was conducted in recent years to provide references for the further extraction of chemical components in cranberry and rational clinical application, which can help to guide people to rationalize their diets and promote the formation of healthy diets. I have the following comments:

1, The present title is a bit convoluted; please optimize.

2, For the Abstract, the full name of the abbreviations CNKI should be provided.

3, The section of Materials and Methods should be removed from the Review.

4, Titles of each subsection in the main body should be a conclusive sentence rather than a simple word.

5, For Table 1, dw and dm should be explained.

6, For Figures 6, 7, 8, and 9, the mechanism should be briefly explained in the Figure Legend.

7, For Table 2, citations should be provided.

Author Response

Comments 1: The present title is a bit convoluted; please optimize.

Response 1: On the 1 page, 2 line. Based on your suggestion, we have also noticed that there is a problem with the title of the article. Thank you for pointing this out. We agree with this comment. Therefore, We modified the title of the article, changing the title “Multi-dimensional study of cranberry: comprehensive evaluation of phytochemical composition, pharmacological mechanism of action, clinical application value, and nutritional significance” to “Cranberry Research Progress: A Systematic Review of Chemical Composition, Pharmacological Mechanisms, Clinical Applications, and Nutritional Significance”. And we have already highlighted it in red in the revised draft.

Comments 2: For the Abstract, the full name of the abbreviations CNKI should be provided.

Response 2: On the 1 page, 29 line. Thank you for your suggestions and for pointing out our shortcomings. We added the full name of the system (China National Knowledge Infrastructure) after CNKI in the abstract. And we have already highlighted it in red in the revised draft.

Comments 3: The section of Materials and Methods should be removed from the Review.

Response 3: On the 2 page, 85 line. Thank you very much for your suggestion. This section plays a crucial role in the text. The “Materials and Methods” section allows readers to understand the research methodology employed in this review, thereby guiding their reading. Therefore, we believe this section should be retained.

Comments 4: Titles of each subsection in the main body should be a conclusive sentence rather than a simple word.

Response 4: Thank you very much for your suggestions. The titles of each subsection in the main text are summaries of that section, and each section is a review based on the subtitle. If necessary modifications are required, we will make corrections. Thank you again for your suggestions.

Comments 5: For Table 1, dw and dm should be explained.

Response 5: On the 9 page, 232 line. Thank you very much for your question. In the article, dw refers to dry weight and dm refers to dry matter, both of which indicate the content of different phenolic acid chemical components in dried cranberries. In addition, we added captions to the tables in the article and highlighted them in red.

Comments 6: For Figures 6, 7, 8, and 9, the mechanism should be briefly explained in the Figure Legend.

Response 6:Thank you very much for your suggestions regarding the images in the article. We have added a brief overview to the captions in accordance with your comments. On the 12 page, 321 line, We added “ The three active components of cranberries work synergistically through multiple pathways, including ‘antioxidant (Nrf2) + metabolic regulation (AMPK/PI3K) + inflammation inhibition (MAPK),’ ultimately enhancing cellular antioxidant capacity, improving metabolism, and inhibiting inflammation.” On the 13 page, 359 line, We added“ Cranberry proanthocyanidin A2 exerts its effects through the pathway of ‘inhibiting DNA gyrase → disrupting nucleic acid metabolism → inhibiting gene expression and cell proliferation.’” On the 14 page, 385 line, We added “Cyanidinol in cranberries precisely inhibits the ‘inflammation-destruction’ cycle of periodontitis through a cascade effect of ‘receptor binding → autophagy regulation → immune polarization → tissue regeneration,’ while simultaneously initiating the repair process.” On the 15 page, 413 line, We added “Quercetin forms a synergistic network through three dimensions: mitochondrial apoptosis, proliferation signal inhibition, and cycle arrest, ultimately achieving the effect of inhibiting cell survival and proliferation.”

Comments 7:For Table 2, citations should be provided.

Response 7:Thank you very much for your suggestions. Regarding the reference to Table 2, we mentioned on page 18, line 532 of the article that the table information was collected from the USDA, and we also mentioned the website where the information was sourced.

Dear Editors and Reviewers,

Wishing you all the best in your endeavors. Thank you for taking the time to review our paper amidst your busy schedule, and for your professional suggestions. Those comments are all valuable and very helpful for revising and improving our paper, as well as the important guiding significance to our researches. In the following, we have provided detailed responses to reviewers' comments. We have tried our best to make all the revisions clear, and we hope that the revised manuscript meets the requirements for publication.

When submitting the revised draft, we provided two documents: one without red markups and another with red markups included in the supplementary files for your review.

Sincerely,

Jiao Xu

Jiamusi College, Heilongjiang University of Traditional Chinese Medicine,

Jiamusi, China

Email: xujiao2023@163.com

Reviewer 2 Report

Comments and Suggestions for Authors

I appreciate the opportunity to review this article. The topic is of interest and relevant to the field. However, there are several aspects that require clarification and revision to improve the overall quality and scientific rigour of the paper. Find below some detailed suggestions.

Title:

  1. The title should be revised, as the current wording “Multi-dimensional study of cranberry” does not clearly indicate the study design. Consider specifying that this is a narrative/comprehensive review.

First page and abstract:

  1. The author’s surname Xiao should be presented in uppercase (Lijin Xiao) in accordance with the formatting requirements.
  2. At line 28, the typographical error “Scoups” should be corrected to “Scopus”. In the same line, the abbreviation “CNCK” should be written in full, as this is its first occurrence of this term in the manuscript.
  3. At line 28, it is also stated that the study used information from some databases and “related papers”; however, this sentence requires clarification. From which source were these additional related papers obtained?

Material and methods:

  1. At line 87, the typographical error “Scoups” should be corrected to “Scopus”.
  2. This manuscript is based on a review; however, the methodology applied for studies selection was not mentioned.
  3. At line 89, the word “al” appears incorrectly and does not fit within the sentence. Please review and correct this usage.
  4. At line 89, “other sources” of data are mentioned. However, this expression is vague. It is important to specify exactly which sources were used and how they contributed to the study, in order to keep the reproducibility of the research.
  5. At line 91, all the software mentioned should be accompanied by the respective version numbers, to ensure clarity and reproducibility of the study. In addition, the word “etc” should be replaced with the correct type of data that was obtained by this software.

Cranberry chemical composition section:

  1. At lines 97–98, calcium, potassium, and magnesium are incorrectly classified as “vitamins”. These are essential minerals, the correction is required to ensure scientific accuracy.
  2. At lines 121–124, the sentences are repetitive and lack clarity. I recommend rewriting this section to improve readability and ensure a more concise presentation of the information.
  3. At line 127, the phrase “Compared with the traditional solvent method” is used. However, it is not clear which specific method is being referred to.
  4. In Figure 1, the term “procyanidins” is used; however, this term has not been previously introduced in the text, where it was referred to as “type A proanthocyanidins”. If this specific subclass is intended, please ensure consistent terminology across the figure label, caption, and main text to avoid confusion.
  5. Figure 1 presents the major proanthocyanidin-like chemical constituents in cranberry. However, this figure is not referenced or discussed within the main text and is only mentioned in parentheses at the end of the “3.1 Proanthocyanidins” section. The same issue occurs with Figure 2, Figure 3, Figure 4 and Figure 5. It is not properly referenced in the main text. Please ensure that all figures are clearly cited and integrated into the relevant sections to guide the reader.
  6. There appear to be two different sections with the same title, “Proanthocyanidins” (sections 3.1 and 3.2, lines 113 and 140). This may confuse the reader.
  7. At line 171, the word “etc.” is used. This should be replaced with a complete list of all flavonol components identified in cranberry, as providing these details is central to the focus of the manuscript.
  8. At line 179, an impact on the physiological activity of cranberries is mentioned; however, this statement is vague. The entire sentence (lines 176-179) should be reformulated to clearly specify the observed effects.
  9. The title of section 3.6 and 4.5, “Other,” are vague. It is recommended to replace it with a more descriptive title to clearly indicate the content of the section.

Cranberry pharmacology section:

  1. First paragraph of section 4, “Cranberry Pharmacology,” should be further developed. It refers to only three studies (references 63-65), despite mentioning multiple effects such as anti-inflammatory, antioxidant, antibacterial, and antitumor activities (as presented in Figure 5). Additionally, Figure 5 should be properly cited and discussed within the main text to guide the reader. Chemical formulas included in the figure should be presented consistently in a single orientation. Currently, they are displayed in multiple directions, which may confuse the reader. Please standardise the presentation for clarity.
  2. Nitric oxide (“NO”) levels are cited at line 282. However, this parameter has not been introduced or defined earlier in the text.
  3. Figure 6 would benefit from a detailed caption or legend explaining all the cited mechanisms of action for each component. This figure again is not properly referenced in the main text. The same occurs in Figure 7, 8 and 9.
  4. Oral health is mentioned at line 327. However, this topic is not further explored. The section currently focuses exclusively on the prevention of urinary tract infections through antimicrobial activity. Consider expanding the discussion to include the effects of cranberry on oral health.
  5. Figure 8 should be revised. The tooth depicted with periodontitis is shown, after treatment, as having regained bone and gingival tissue lost during periodontal destruction, resembling a healthy tooth. This does not accurately reflect the natural healing process, as complete regeneration of lost periodontal structures is unlikely. Please adjust the figure to better represent feasible treatment outcomes. Caption or legends for this figure are also required.

Clinical application of cranberries section:

  1. Item 5, “Clinical Applications of Cranberries,” should be reorganized. Currently, it appears at the end of the manuscript and presents information that is highly repetitive, such as the use of cranberry for urinary tract prevention, which has already been discussed throughout the text. Consider integrating or restructuring this section to avoid redundancy and improve the logical flow.
  2. The phrase “hairy E. coli” (line 416) is not scientifically appropriate. It likely refers to E. coli expressing fimbriae or pili, which are hair-like appendages that enable bacterial adhesion. Please replace “hairy E. coli” with a more precise term, such as “fimbriae-expressing E. coli” or “piliated E. coli,” to ensure clarity and accuracy.
  3. The term “pus” (line 418) is not considered scientific language. It is recommended to use a more precise term, such as “purulent exudate” or “suppurative material,” to describe signs and symptoms.
  4. The studies on oral health are still preliminary and require properly designed clinical trials with robust methodologies. It is not yet possible to state with certainty that cranberry has definitive effects on oral health. While it appears to have potential benefits, this should be clearly communicated in the “Oral Health” section (5.2) to avoid overstating the evidence.
  5. Table 2 seems to be incomplete for certain nutrients, including carbohydrates, sugars, sucrose, and vitamin C. Please review, provide the missing data and refer to the table accordingly within the main text.

Conclusion:

  1. The conclusion should be more concise, highlighting the main findings and the limitations of the study. Phrases such as “it is believed” should be avoided, given that the analysis is based on current data.

Throughout the main text:

The aim of the study is highlighted in the end of several sections, such as “Introduction, Material and methods, Cranberry chemical composition”, When it should be stated only in the introduction and/or material and methods sections.

The text is highly repetitive, particularly at the beginning of each section, where similar introductory statements about cranberry are repeated. This should be revised to reduce redundancy and provide a more concise and engaging presentation of the material.

Finally, the manuscript would benefit from thorough English language editing to improve grammatical accuracy and clarity. A thorough review of spacing and formatting throughout the manuscript is also necessary, as several inconsistencies and errors have been identified (e.g. the use of italics, bold, and other typographical elements).

I appreciate the opportunity to review this manuscript. Hopefully, this suggestion will help enhance clarity and scientific quality of your manuscript.

Author Response

Comments 1: The title should be revised, as the current wording “Multi-dimensional study of cranberry” does not clearly indicate the study design. Consider specifying that this is a narrative/comprehensive review.

Response 1: Thank you very much for your suggestions on our article. We welcome your feedback. We have revised the title of our article and changed it to “Cranberry Research Progress: A Systematic Review of Chemical Composition, Pharmacological Mechanisms, Clinical Applications, and Nutritional Significance”. I hope that the revised question meets your requirements.

Comments 2: The author’s surname Xiao should be presented in uppercase (Lijin Xiao) in accordance with the formatting requirements.

Response 2: We sincerely apologize for this mistake caused by our carelessness. As per your request, we have changed “Lijin xiao” to “Lijin Xiao”. And highlighted in red in the article.

Comments 3: At line 28, the typographical error “Scoups” should be corrected to “Scopus”. In the same line, the abbreviation “CNCK” should be written in full, as this is its first occurrence of this term in the manuscript.

Response 3: On the 1 page, 29 line. Thank you for your suggestions and for pointing out our shortcomings. We added the full name of the system (China National Knowledge Infrastructure) after CNKI in the abstract. And we have already highlighted it in red in the revised draft.

Comments 4: At line 28, it is also stated that the study used information from some databases and “related papers”; however, this sentence requires clarification. From which source were these additional related papers obtained?

Response 4: Thank you very much for your question. We will explain your question. The papers and databases reviewed in our article are sourced from PubMed, Web of Science, Scopus, and CNKI. All references are drawn from these databases and are introduced in the introduction.

Comments 5: At line 87, the typographical error “Scoups” should be corrected to “Scopus”.

Response 5: Thank you very much for your tolerance of our mistake. On the 2 page, line 89, we have corrected our mistake and highlighted it in red in the text.

Comments 6: This manuscript is based on a review; however, the methodology applied for studies selection was not mentioned.

Response 6: Thank you for your inquiry regarding this section. The methodology for this article review is outlined in Section 2, “Materials and Methods,” which includes the search portals and graphing software utilized during the process.

Comments 7: At line 89, the word “al” appears incorrectly and does not fit within the sentence. Please review and correct this usage.

Response 7: Thank you very much for your question. On line 92, The “al” shown in the article is due to a paragraph formatting issue, which caused the letters in “pharmacological” to shift. We have already corrected this issue in the article.

Comments 8: At line 89, “other sources” of data are mentioned. However, this expression is vague. It is important to specify exactly which sources were used and how they contributed to the study, in order to keep the reproducibility of the research.

Response 8: Thank you very much for your advice. On page 2, line 90, we made changes to the article, adding other data sources, including “USDA FoodData Central, EFSA, CFIA,” and highlighted them in red in the text.

Comments 9: At line 91, all the software mentioned should be accompanied by the respective version numbers, to ensure clarity and reproducibility of the study. In addition, the word “etc” should be replaced with the correct type of data that was obtained by this software.

Response 9: Thank you very much for your detailed suggestions on our article. Based on your suggestions, we have made modifications to the article on page 3, line 93, adding the version number of the software used in the writing process.

Comments 10: At lines 97–98, calcium, potassium, and magnesium are incorrectly classified as “vitamins”. These are essential minerals, the correction is required to ensure scientific accuracy.

Response 10: Thank you very much for reviewing the article. We apologize for our mistake. On page 3, line 98, we have changed “vitamin” to “mineral” as you requested.

Comments 11: At lines 121–124, the sentences are repetitive and lack clarity. I recommend rewriting this section to improve readability and ensure a more concise presentation of the information.

Response 11: Based on your suggestion, we have corrected this part of the narrative and modified the language, changing “Due to its richness in type A proanthocyanidins, cranberry can effectively inhibit the adhesion of Escherichia coli to the uroepithelial cells [28]. Type A proanthocyanidins, as the signature components of cranberry distinguishing it from other berries, demonstrated its unique anti-urinary tract infection activity [29].” to “Owing to its abundance of type A proanthocyanidins, cranberry can effectively impede the adhesion of Escherichia coli to uroepithelial cells [28]. Type A proanthocyanidins, serving as the characteristic components of cranberry that differentiate it from other berries, have manifested their distinctive anti - urinary tract infection activity [29].”

Comments 12: At line 127, the phrase “Compared with the traditional solvent method” is used. However, it is not clear which specific method is being referred to.

Response 12: The “Compared with the traditional solvent method” mentioned in the article, this refers to a mixture separation method, also known as coprecipitation, which involves dissolving the target substance and carrier in an organic solvent and then evaporating the solvent to form a coprecipitate. Commonly used carrier materials include PVP, galactose, mannose, and bile acids. This method is suitable for the separation of thermally unstable or volatile components.

Comments 13: In Figure 1, the term “procyanidins” is used; however, this term has not been previously introduced in the text, where it was referred to as “type A proanthocyanidins”. If this specific subclass is intended, please ensure consistent terminology across the figure label, caption, and main text to avoid confusion.

Response 13: Thank you very much for your suggestions on the details of the article. In Figure 1, we have drawn the proanthocyanidins, including their monomers and the type A proanthocyanidins extracted from cranberries. These are included in the type A proanthocyanidins mentioned in the text, meaning that the proanthocyanidins included in the figure are more comprehensive.

Comments 14: Figure 1 presents the major proanthocyanidin-like chemical constituents in cranberry. However, this figure is not referenced or discussed within the main text and is only mentioned in parentheses at the end of the “3.1 Proanthocyanidins” section. The same issue occurs with Figure 2, Figure 3, Figure 4 and Figure 5. It is not properly referenced in the main text. Please ensure that all figures are clearly cited and integrated into the relevant sections to guide the reader.

Response 14: Thank you very much for your feedback. We understand your suggestions and have made revisions to the article. The specific revisions are highlighted in red in the text.

Comments 15: There appear to be two different sections with the same title, “Proanthocyanidins” (sections 3.1 and 3.2, lines 113 and 140). This may confuse the reader.

Response 15: Thank you very much for your careful review. We have also noted our mistake. We have revised section 3 “3.2 Proanthocyanidins” to read “3.2 Anthocyanin.”

Comments 16: At line 171, the word “etc.” is used. This should be replaced with a complete list of all flavonol components identified in cranberry, as providing these details is central to the focus of the manuscript.

Response 16: Thank you very much for your reminder. On line 176, we changed “etc.” to “myricetin, myricetin 3-O-galactoside, hyperpside, isoquercitrin, quercetin 3-O-α-L-arabinoside” in the article. We have revised the description of flavonoid compounds in the article based on your comments.

Comments 17: At line 179, an impact on the physiological activity of cranberries is mentioned; however, this statement is vague. The entire sentence (lines 176-179) should be reformulated to clearly specify the observed effects.

Response 17: Thank you very much for your suggestions regarding the completeness of the article content. We have made revisions based on your feedback. We have expanded the content in lines 185 to enrich the impact of changes in chemical composition on the physiological activity of cranberries, and these impacts directly affect their clinical application. We have highlighted the modified content in red.

Comments 18: The title of section 3.6 and 4.5, “Other,” are vague. It is recommended to replace it with a more descriptive title to clearly indicate the content of the section.

Response 18: Thank you very much for your advice. In lines 262 and 416 of the article, we changed the subheadings to “Other Compounds in Cranberry” and “Other Pharmacological Effects of Cranberry,” respectively.

Comments 19: First paragraph of section 4, “Cranberry Pharmacology,” should be further developed. It refers to only three studies (references 63-65), despite mentioning multiple effects such as anti-inflammatory, antioxidant, antibacterial, and antitumor activities (as presented in Figure 5). Additionally, Figure 5 should be properly cited and discussed within the main text to guide the reader. Chemical formulas included in the figure should be presented consistently in a single orientation. Currently, they are displayed in multiple directions, which may confuse the reader. Please standardise the presentation for clarity.

Response 19: Thank you very much for your valuable suggestions. We have made revisions based on your comments and standardized the orientation of the chemical formulas in Figure 5. In addition, we have expanded on this paragraph.

Comments 20: Nitric oxide (“NO”) levels are cited at line 282. However, this parameter has not been introduced or defined earlier in the text.

Response 20: Thank you very much for your reminder. On line 302, we inserted an introduction to “NO,” which serves as a stepping stone for the appearance of “NO” in the next question, enriching the paragraph.

Comments 21: Figure 6 would benefit from a detailed caption or legend explaining all the cited mechanisms of action for each component. This figure again is not properly referenced in the main text. The same occurs in Figure 7, 8 and 9.

Response 21: Thank you very much for your suggestions regarding the images in the article. We have added a brief overview to the captions in accordance with your comments. On the 12 page, 321 line, We added “ The three active components of cranberries work synergistically through multiple pathways, including ‘antioxidant (Nrf2) + metabolic regulation (AMPK/PI3K) + inflammation inhibition (MAPK),’ ultimately enhancing cellular antioxidant capacity, improving metabolism, and inhibiting inflammation.” On the 13 page, 359 line, We added“ Cranberry proanthocyanidin A2 exerts its effects through the pathway of ‘inhibiting DNA gyrase → disrupting nucleic acid metabolism → inhibiting gene expression and cell proliferation.’” On the 14 page, 385 line, We added “Cyanidinol in cranberries precisely inhibits the ‘inflammation-destruction’ cycle of periodontitis through a cascade effect of ‘receptor binding → autophagy regulation → immune polarization → tissue regeneration,’ while simultaneously initiating the repair process.” On the 15 page, 413 line, We added “Quercetin forms a synergistic network through three dimensions: mitochondrial apoptosis, proliferation signal inhibition, and cycle arrest, ultimately achieving the effect of inhibiting cell survival and proliferation.”

Comments 22: Oral health is mentioned at line 327. However, this topic is not further explored. The section currently focuses exclusively on the prevention of urinary tract infections through antimicrobial activity. Consider expanding the discussion to include the effects of cranberry on oral health.

Response 22: Thank you for your suggestion. In this paragraph, we added information about cranberries' ability to inhibit certain bacteria that cause oral health problems. The main purpose of this paragraph is to explain that cranberries can inhibit bacterial growth, thereby exerting an antibacterial effect and treating certain bacterial infections.

Comments 23: Figure 8 should be revised. The tooth depicted with periodontitis is shown, after treatment, as having regained bone and gingival tissue lost during periodontal destruction, resembling a healthy tooth. This does not accurately reflect the natural healing process, as complete regeneration of lost periodontal structures is unlikely. Please adjust the figure to better represent feasible treatment outcomes. Caption or legends for this figure are also required.

Response 23: Based on your professional advice, we have modified the changes in periodontitis after cranberry treatment in Figure 8. The modified results are shown in Figure 8 of the manuscript.

Comments 24: Item 5, “Clinical Applications of Cranberries,” should be reorganized. Currently, it appears at the end of the manuscript and presents information that is highly repetitive, such as the use of cranberry for urinary tract prevention, which has already been discussed throughout the text. Consider integrating or restructuring this section to avoid redundancy and improve the logical flow.

Response 24: Thank you very much for your suggestion. We have revised the relevant section, and the revised content is highlighted in red in the article.

Comments 25: The phrase “hairy E. coli” (line 416) is not scientifically appropriate. It likely refers to E. coli expressing fimbriae or pili, which are hair-like appendages that enable bacterial adhesion. Please replace “hairy E. coli” with a more precise term, such as “fimbriae-expressing E. coli” or “piliated E. coli,” to ensure clarity and accuracy.

Response 25: Thank you very much for your professional advice. We will follow your recommendations. On line 449, we changed “hairy E. coli” to “fimbriae-expressing E. coli.”

Comments 26: The term “pus” (line 418) is not considered scientific language. It is recommended to use a more precise term, such as “purulent exudate” or “suppurative material,” to describe signs and symptoms.

Response 26: Thank you very much for your advice. In line 446, we changed “pus” to “purulent exudate.”

Comments 27: The studies on oral health are still preliminary and require properly designed clinical trials with robust methodologies. It is not yet possible to state with certainty that cranberry has definitive effects on oral health. While it appears to have potential benefits, this should be clearly communicated in the “Oral Health” section (5.2) to avoid overstating the evidence.

Response 27: Thank you very much for your careful review of our article. We apologize for not considering this point thoroughly when writing the article. We have added the note “Current experimental studies have only preliminarily verified that cranberries have therapeutic effects on periodontitis, dental caries, and other oral infectious diseases, but it is still unclear whether cranberries have such therapeutic effects in clinical treatment. Therefore, more experiments are needed to prove the therapeutic effects of cranberries in clinical practice.” in line 492 to eliminate any confusion on the part of readers.

Comments 28: Table 2 seems to be incomplete for certain nutrients, including carbohydrates, sugars, sucrose, and vitamin C. Please review, provide the missing data and refer to the table accordingly within the main text.

Response 28: Thank you very much for your careful review. We have expanded the table and referenced it in the main text.

Comments 29: The conclusion should be more concise, highlighting the main findings and the limitations of the study. Phrases such as “it is believed” should be avoided, given that the analysis is based on current data.

Response 29: Thanks for the suggestion. We revised Section 7 of the article to simplify it.

Dear Editors and Reviewers,

Wishing you all the best in your endeavors. Thank you for taking the time to review our paper amidst your busy schedule, and for your professional suggestions. Those comments are all valuable and very helpful for revising and improving our paper, as well as the important guiding significance to our researches. In the following, we have provided detailed responses to reviewers' comments. We have tried our best to make all the revisions clear, and we hope that the revised manuscript meets the requirements for publication.

When submitting the revised draft, we provided two documents: one without red markups and another with red markups included in the supplementary files for your review.

Sincerely,

Jiao Xu

  Jiamusi College, Heilongjiang University of Traditional Chinese Medicine,

Jiamusi, China

Email: xujiao2023@163.com

Reviewer 3 Report

Comments and Suggestions for Authors

The review article, as the title suggests, covers a comprehensive assessment of the phytochemical composition, pharmacological mechanism of action, value for clinical use, and nutritional value of cranberries. However, it is not entirely clear which cranberries, Vaccinium macrocarpon or the genus Vaccinium? The authors should clarify this, especially in the section on chemical composition. The positive aspect of the article is the large coverage of multilateral information, however, it requires serious revision before publication. In particular, I list them below in the text of the review.

Line 13 Vaccinium macrocarpon, Line 14 Vaccinium – Latin names should be written in italics here and below

Line 14 Vaccinium macrocarpon belongs to the Ericaceae family, if the authors have any new data, please provide a link or correct the error.

Lines 15-16 “Cranberries are mainly distributed in the northern United States, China's Daxinganling region, China's Heilongjiang Province, Fuyuan City”. This probably means that the plant is cultivated in China's Heilongjiang Province, then it should be mentioned where it was naturalized, for example, in Europe. Otherwise, it creates a false impression that it always grows in China, as in North America, and this is not true.

Lines 15-17. “Cranberries are rich in a variety of biologically active components, such as polyphenols (proanthocyanidins, chlorogenic acid, flavonols, anthocyanins, caffeic acid, etc.), triterpenoids, chlorogenic acid and vitamins and other nutrients.” The enumeration of secondary metabolites of cranberries should be brought into a single classification, for example, in the authors' text chlorogenic acid is mentioned twice, vitamins are highlighted separately, but what about vitamin A related to terpenoids?

Limes 38-40 “Cranberry (Vaccinium macrocarpon Ait.) is an herbaceous plant of thegenus Vaccinium in the family Rhododendron, an evergreen shrub with roundedorobovate leaves; white or reddish, campanulate flowers, and purplish-red berries”. Here, the authors should refer to the classical botanical work in order to exclude errors.

There are a lot of errors in the introduction of the article, both grammatical and scientific. The text is poorly written in terms of the sequence of presentation, information about the usefulness of cranberries and its substances on the urinary system is repeated many times. It is necessary to make the presentation of the material more consistent. The introduction requires a complete revision of the text.

Line 87 “Scoups” typo in the text

Lines 97-98 “…vitamins (e.g., calcium, potassium, and magnesium), vitamin C, and vitamin E”. The inaccuracy of the translation probably refers to minerals.

When describing the compounds found in Vaccinium macrocarpon, reference should be made to primary sources. And a table is needed, it is possible in the form of additional materials. Broken down by groups of secondary metabolites, individual compounds detected, isolation methods, concentration, and information source.

The article lacks a systematic presentation of the material. I recommend that the data be presented in tabular form, dividing the table into systematic groups of secondary metabolites.

Table 1. Biological activities of secondary metabolites of Vaccinium macrocarpon.

Activity

Pure Substance/Extract

Model

Mechanism of Action

References

Well-known facts about compounds and the biological activity of well-known individual compounds, such as quercetin, should be excluded from the text of the article.  To focus on the unique groups of secondary metabolites and individual compounds characteristic of cranberries. To disclose information about their biological activity and to describe the mechanism of action in an accessible way and to what extent this mechanism has been proven.

The methods for isolating compounds are interesting, but they should be described for all compounds and highlighted in a separate section.

Comments on the Quality of English Language

The English needs major revisions.

Author Response

Comments 1: Line 13 Vaccinium macrocarpon, Line 14 Vaccinium – Latin names should be written in italics here and below.

Response 1: Thank you very much for reading our article carefully. We apologize for the errors in the details of the article. We have corrected all italicized Latin names mentioned in the article.

Comments 2: Line 14 Vaccinium macrocarpon belongs to the Ericaceae family, if the authors have any new data, please provide a link or correct the error.

Response 2: We sincerely appreciate your patience and understanding regarding our error. We have made the necessary corrections. On line 13, we changed “Rhododendron” to “Ericaceae”.

Comments 3: Lines 15-16 “Cranberries are mainly distributed in the northern United States, China's Daxinganling region, China's Heilongjiang Province, Fuyuan City”. This probably means that the plant is cultivated in China's Heilongjiang Province, then it should be mentioned where it was naturalized, for example, in Europe. Otherwise, it creates a false impression that it always grows in China, as in North America, and this is not true.

Response 3: On line 13, Thank you very much for your suggestion. We have modified the cranberry growing region as per your request. And highlight it in red within the article.

Comments 4: Lines 15-17. “Cranberries are rich in a variety of biologically active components, such as polyphenols (proanthocyanidins, chlorogenic acid, flavonols, anthocyanins, caffeic acid, etc.), triterpenoids, chlorogenic acid and vitamins and other nutrients.” The enumeration of secondary metabolites of cranberries should be brought into a single classification, for example, in the authors' text chlorogenic acid is mentioned twice, vitamins are highlighted separately, but what about vitamin A related to terpenoids?

Response 4: Thank you very much for your professional advice. On line 19, we have revised the statement “Cranberries are rich in a variety of biologically active components, such as polyphenols (proanthocyanidins, chlorogenic acid, flavonols, anthocyanins, caffeic acid, etc.), triterpenoids, chlorogenic acid, and vitamins and other nutrients.” Revised to: “Cranberries are rich in a variety of biologically active components, such as polyphenols (proanthocyanidins, chlorogenic acid, flavonols, anthocyanins, caffeic acid, etc.), triterpenoids, and other nutrients.”

Comments 5: Limes 38-40 “Cranberry (Vaccinium macrocarpon Ait.) is an herbaceous plant of thegenus Vaccinium in the family Rhododendron, an evergreen shrub with roundedorobovate leaves; white or reddish, campanulate flowers, and purplish-red berries”. Here, the authors should refer to the classical botanical work in order to exclude errors.

Response 5: On line 39-41, After reviewing the materials, we revised the content of the sentence to enhance its technical accuracy, replacing “Cranberry (Vaccinium macrocarpon Ait.) is an herbaceous plant of the genus Vaccinium in the family Rhododendron, an evergreen shrub with rounded or obovate leaves; white or reddish, campanulate flowers, and purplish-red berries ” with “Cranberry (Vaccinium macrocarpon Ait.) belongs to the genus Vaccinium within the family Ericaceae. It is an evergreen shrub characterized by rounded or obovate leaves, white or reddish campanulate flowers, and purplish - red berries.”

Comments 6: Line 87 “Scoups” typo in the text.

Response 6: We sincerely apologize for our mistake. We have corrected the error by replacing “Scoups” with “Scopus”. Additionally, the error on line 89 has been corrected.

Comments 7: Lines 97-98 “…vitamins (e.g., calcium, potassium, and magnesium), vitamin C, and vitamin E”. The inaccuracy of the translation probably refers to minerals.

Response 7: Thank you very much for your suggestion. We have made the necessary revisions in line 98 to clarify the distinction between minerals and vitamins.

Comments 8: When describing the compounds found in Vaccinium macrocarpon, reference should be made to primary sources. And a table is needed, it is possible in the form of additional materials. Broken down by groups of secondary metabolites, individual compounds detected, isolation methods, concentration, and information source.

Response 8: Thank you very much for your suggestions. We have added a table to the section on compounds in the text, detailing the content of compounds in cranberries. We hope this table will be helpful to the article.

Dear Editors and Reviewers,

Wishing you all the best in your endeavors. Thank you for taking the time to review our paper amidst your busy schedule, and for your professional suggestions. Those comments are all valuable and very helpful for revising and improving our paper, as well as the important guiding significance to our researches. In the following, we have provided detailed responses to reviewers' comments. We have tried our best to make all the revisions clear, and we hope that the revised manuscript meets the requirements for publication.

When submitting the revised draft, we provided two documents: one without red markups and another with red markups included in the supplementary files for your review.

Sincerely,

Jiao Xu

  Jiamusi College, Heilongjiang University of Traditional Chinese Medicine,

Jiamusi, China

Email: xujiao2023@163.com

Round 2

Reviewer 1 Report

Comments and Suggestions for Authors

Authors have addressed my concerns in the revision.

Author Response

Dear Editors and Reviewers,

Thank you for your valuable feedback, which has significantly enhanced the readability and scientific rigor of the revised manuscript. Wishing you all the best in your endeavors and smooth sailing in all matters.

Sincerely,

Jiao Xu

  Jiamusi College, Heilongjiang University of Traditional Chinese Medicine,

Jiamusi, China

Email: xujiao2023@163.com

Reviewer 2 Report

Comments and Suggestions for Authors

Journal: International Journal of Molecular Sciences

Article: Multi-dimensional study of cranberry: comprehensive evaluation of phytochemical composition, pharmacological mechanism of action, clinical application value, and nutritional significance

Authors: Jiao Xu, Chenliang Li, Wei Wu, Lingyang Kong, Lijin Xiao, Wei Ma, Lihong Zhang

I appreciate the authors’ modifications and the improvements made in the revised manuscript. Overall, the topic is of potential interest; however, there are significant issues regarding the study design, clarity of objectives, and methodological rigor that need to be addressed. The following comments are intended to provide constructive guidance to help improve the manuscript for a future submission.

Title:

  1. The title has been modified; however, the manuscript now appears to present a different study design. Initially (in the first review), it was described as a narrative/comprehensive review, but it is currently presented as a systematic review. Please note that a systematic review is a considerably more complex study type and requires a clearly structured and transparent methodology, which is not adequately demonstrated in the current version of this manuscript.

Major issues:

  1. The requirements for a systematic review include, among others: a well-defined research question (e.g., structured with PICO components), a detailed and registered research protocol (based on validated guidelines), a comprehensive and reproducible search strategy across the databases, rigorous inclusion and exclusion criteria, detailed description of study selection and data selection processes, assessment of the methodological quality of the included studies, a transparent specification and presentation of extracted data (e.g., tables with authors and study characteristics), and an up-to-date synthesis of the findings. Unfortunately, these essential elements are not demonstrated in the current manuscript, and therefore it cannot be considered a systematic review in its present form.

  1. The current version of the manuscript represents a different study from the one that was submitted to the journal. In its present form, it should be considered a new submission rather than a revision of the original manuscript.

Additional comments:

  1. Figure 1 is not cited in the text, and at line 124 the reference to Figure 2 appears to be incorrect. Please revise the figure citations to ensure accuracy and consistency throughout the manuscript.
  2. The subtitle "1.5 Others" is unclear and does not convey meaningful information. Consider providing a more descriptive heading that accurately reflects the content of this section.
  3. According to Table 1, out of 106 analyzed papers, only references 17 and 19 report data on cranberry anthocyanin and flavonols compounds. Could the authors clarify whether this accurately reflects the focus of the review?
  4. Vaccinium macrocarpon belongs to the Ericaceae family, not the Rhododendron family as mentioned in line 326 and elsewhere in the manuscript. Please correct this taxonomic error throughout the text.

Throughout the main text:

The aim of the study is unclear, as the manuscript does not present a specific research question to be addressed, which is essential for a systematic review. The authors should clarify the objective and frame it around a clear and answerable question.

The manuscript would benefit from thorough English language editing to improve grammatical accuracy and clarity. A thorough review of spacing and formatting throughout the manuscript is also necessary, as several inconsistencies and typographical errors have been identified.

Thank you for considering the submission. Hopefully, the comments provided will be helpful in refining the clarity and scientific quality of your manuscript. With a rigorous approach and careful adherence to methodological standards, your work could make a valuable contribution in a future submission.

Author Response

Comments 1: The title has been modified; however, the manuscript now appears to present a different study design. Initially (in the first review), it was described as a narrative/comprehensive review, but it is currently presented as a systematic review. Please note that a systematic review is a considerably more complex study type and requires a clearly structured and transparent methodology, which is not adequately demonstrated in the current version of this manuscript.

Response 1: Thank you very much for your guidance. On page 1, line 2, we changed “system review” in the article title to “review.” We sincerely apologize for the errors in the article resulting from our unfamiliarity with the systematic review process during the revision. Thank you again for your valuable suggestions.

Comments 2: Figure 1 is not cited in the text, and at line 124 the reference to Figure 2 appears to be incorrect. Please revise the figure citations to ensure accuracy and consistency throughout the manuscript.

Response 2: On page 3, line 120, We have revised the citation in Figure 1, changing (Figure 2) to (Figure 1). We sincerely apologize for our oversight during the revision process and thank you again for your suggestion.

Comments 3: The subtitle "1.5 Others" is unclear and does not convey meaningful information. Consider providing a more descriptive heading that accurately reflects the content of this section.

Response 3: Thank you very much for your suggestions. On page 3, line 121, we revised “2.5 Others” to “2.5 Other research methods” to reduce reader confusion and guide their reading.

Comments 4: According to Table 1, out of 106 analyzed papers, only references 17 and 19 report data on cranberry anthocyanin and flavonols compounds. Could the authors clarify whether this accurately reflects the focus of the review?

Response 4: In the discussion of cranberry proanthocyanidins and flavanol compounds, 106 publications provided varying degrees of data on the content of these two categories of components. The data presented in these two publications were more comprehensive and thus more valuable for reference. Therefore, we have prioritized citing these two publications in the table.

Comments 5: Vaccinium macrocarpon belongs to the Ericaceae family, not the Rhododendron family as mentioned in line 326 and elsewhere in the manuscript. Please correct this taxonomic error throughout the text.

Response 5: On page 13, line 320, we revised “Rhododendron” to “Ericaceae”. We sincerely appreciate your careful review of our paper and thank you for your understanding regarding our errors.

Dear Editors and Reviewers,

We extend our sincerest wishes for the utmost success in your pursuits. We are profoundly grateful for your meticulous examination of our manuscript amidst your demanding schedule, and for the esteemed recommendations you have proffered. Each of your comments holds significant value and has been instrumental in refining and enhancing our work, as well as serving as a critical guidepost for our research endeavors. Subsequently, we have furnished comprehensive replies to the critiques offered by the reviewers. We have endeavored to ensure that all modifications are transparently presented, and we trust that the revised version of our manuscript now fulfills the criteria for publication.

Sincerely,

Jiao Xu

  Jiamusi College, Heilongjiang University of Traditional Chinese Medicine,

Jiamusi, China

Email: xujiao2023@163.com

Reviewer 3 Report

Comments and Suggestions for Authors

The authors have significantly improved the manuscript. However, the main question of why the description of the composition is based on the composition of the genus, and not the species under study, remains open. In addition, the source of information 40 from ‘Table 3. Major phenolic acid chemical constituents in cranberries’ is about a different species altogether - Bojilov, D., et al., HPLC Analysis and In Vitro and In Silico Evaluation of the Biological Activity of Polyphenolic Components Separated with Solvents of Various Polarities from Helichrysum italicum. Molecules, 2023. 28(17): p. 6198. It is necessary to remove such errors throughout the manuscript. The description of the composition should refer only to the original source. And since the article as a whole is about the genus cranberry, remove the name of the species Vaccinium macrocarpon Ait. from the abstract, and rewrite that the review is devoted to the genus cranberry. Minor comments:

Line 326 fix family Rhododendron

Add to figure captions the applications or resources used to create them, BioRender or other (?).

Sincerely,

JSMI Reviewer

Author Response

Comments 1:  The main question of why the description of the composition is based on the composition of the genus, and not the species under study, remains open.

Response 1: Thank you very much for your question. All species research in this article is based on the cranberry (Vaccinium macrocarpon) of the Ericaceae family. The chemical composition and pharmacological effects described herein pertain exclusively to cranberries.

Comments 2: In addition, the source of information 40 from ‘Table 3. Major phenolic acid chemical constituents in cranberries’ is about a different species altogether - Bojilov, D., et al., HPLC Analysis and In Vitro and In Silico Evaluation of the Biological Activity of Polyphenolic Components Separated with Solvents of Various Polarities from Helichrysum italicum. Molecules, 2023. 28(17): p. 6198. It is necessary to remove such errors throughout the manuscript.

Response 2: Thank you very much for bringing this to our attention. We sincerely apologize for overlooking this table during the article review and revision process. We have now correctly modified the references to this table within the text.

Comments 3: Line 326 fix family Rhododendron.

Response 3: On page 13, line 320, we revised “Rhododendron” to “Ericaceae”. We sincerely appreciate your careful review of our paper and thank you for your understanding regarding our errors.

Dear Editors and Reviewers,

We extend our sincerest wishes for the utmost success in your pursuits. We are profoundly grateful for your meticulous examination of our manuscript amidst your demanding schedule, and for the esteemed recommendations you have proffered. Each of your comments holds significant value and has been instrumental in refining and enhancing our work, as well as serving as a critical guidepost for our research endeavors. Subsequently, we have furnished comprehensive replies to the critiques offered by the reviewers. We have endeavored to ensure that all modifications are transparently presented, and we trust that the revised version of our manuscript now fulfills the criteria for publication.

Sincerely,

Jiao Xu

  Jiamusi College, Heilongjiang University of Traditional Chinese Medicine,

Jiamusi, China

Email: xujiao2023@163.com

Round 3

Reviewer 2 Report

Comments and Suggestions for Authors

I appreciate the authors’ efforts in revising the manuscript; however, critical issues remain that require careful correction before the manuscript can be considered for further review.

Material and methods:

  1. At line 121, the reference to Figure 2 is still incorrect. The authors are referring to the screening process and should most likely cite Figure 1, which presents the flowchart of the study. Please correct this in-text citation.

  1. Throughout the manuscript, there are multiple instances of missing spaces between words, for example: “systematicallyassessing” (line 87), “searchespertaining” (line 88), “duplicateentries” (lines 88-89), “abstractsto” (line 89), “havebeen” (line 110), “furtherevaluation” (line 117), and “thisrigorous” (line 120), among others. Please carefully review the entire text to correct all spacing errors, including ensuring proper spacing after commas (, ). These are basic formatting issues that must be addressed consistently throughout the manuscript.

  1. In section 2.3 (line 107), the heading is written as “Inclusion and non-inclusion criteria.” This should be corrected to “Inclusion and exclusion criteria.”

  1. At line 115, the subsection is titled “2.4 Search results.” It is inappropriate to present results within the Methods section. This content should be moved to the Results section and presented as the first result, with the subsection title revised accordingly.

  1. The authors state: “On page 3, line 121, we revised ‘2.5 Others’ to ‘2.5 Other research methods.’” However, this change appears on line 122, not line 121. In addition, the heading “Other research methods” does not accurately reflect the content of the paragraph that follows, as already noted in my previous review. Please revise this subheading to something more appropriate, such as “Illustration methods,” or another term that clearly corresponds to the paragraph content. Furthermore, the databases used in this study are already described in section 2.2 (Search strategy), so there is no need to repeat this information here - please remove it.

  1. The reference to Figure 2 at line 152 is incorrectly placed after the period. Please move the citation so that it appears before the period, in accordance with standard scientific writing conventions.

  1. Why is the aim of the study repeated in the Methods section? The sentence “The aim of this study was to systematically summarize the chemical constituents of cranberry and their extraction methods” (lines 150–152) is already stated at the end of the Introduction. Please rephrase or remove this sentence to avoid unnecessary repetition.

Results

  1. A section titled “3. Results” should be added, and the “Search results” subsection should be placed under this main Results section. All subsequent subsections, such as Cranberry Composition and Cranberry Pharmacology, should then be numbered accordingly under Results. You should not start directly with “3. Cranberry chemical composition” without first presenting the main Results heading.

  1. The authors state that they prioritized citing only two publications in Table 1 (references 17 and 19) because “these two publications were more comprehensive and thus more valuable for reference.” If this approach is maintained, the authors should explicitly mention this choice in the text, explaining the rationale for including only these two references. Alternatively, they should include the other relevant references in the table.
Comments on the Quality of English Language

The text is highly repetitive, particularly at the beginning of each section, where similar introductory statements about cranberry are repeated. This should be revised to reduce redundancy and provide a more concise and engaging presentation of the material.

Overall, while some improvements have been made, several fundamental issues—such as section headings, taxonomic errors, and repeated or misplaced content—remain unresolved and must be addressed before the manuscript can be considered for acceptance.

Author Response

Comments 1: At line 121, the reference to Figure 2 is still incorrect. The authors are referring to the screening process and should most likely cite Figure 1, which presents the flowchart of the study. Please correct this in-text citation.

Response 1: Thank you very much for your suggestions. We have revised such errors in the text.

Comments 2: Throughout the manuscript, there are multiple instances of missing spaces between words, for example: “systematicallyassessing” (line 87), “searchespertaining” (line 88), “duplicateentries” (lines 88-89), “abstractsto” (line 89), “havebeen” (line 110), “furtherevaluation” (line 117), and “thisrigorous” (line 120), among others. Please carefully review the entire text to correct all spacing errors, including ensuring proper spacing after commas (, ). These are basic formatting issues that must be addressed consistently throughout the manuscript.

Response 2: Thank you very much for your careful review. We have already addressed similar issues in the text and reviewed all spacing. Should any further instances arise, please notify us and we will make the necessary revisions.

Comments 3: In section 2.3 (line 107), the heading is written as “Inclusion and non-inclusion criteria.” This should be corrected to “Inclusion and exclusion criteria.”

Response 3: Thank you very much for your reminder. We have revised the title as per your suggestion on page 3, line 103, changing “Inclusion and non-inclusion criteria” to “Inclusion and exclusion criteria.”

Comments 4: At line 115, the subsection is titled “2.4 Search results.” It is inappropriate to present results within the Methods section. This content should be moved to the Results section and presented as the first result, with the subsection title revised accordingly.

Response 4: Thank you very much for your suggestions. We have adjusted the content of the search results to the results section as requested and revised the numbering of the subheadings.

Comments 5: The authors state: “On page 3, line 121, we revised ‘2.5 Others’ to ‘2.5 Other research methods.’” However, this change appears on line 122, not line 121. In addition, the heading “Other research methods” does not accurately reflect the content of the paragraph that follows, as already noted in my previous review. Please revise this subheading to something more appropriate, such as “Illustration methods,” or another term that clearly corresponds to the paragraph content. Furthermore, the databases used in this study are already described in section 2.2 (Search strategy), so there is no need to repeat this information here - please remove it.

Response 5: Thank you very much for your suggestions. On line 111, we changed the heading to “Illustration methods” and removed the duplicate database entries within the paragraph.

Comments 6: The reference to Figure 2 at line 152 is incorrectly placed after the period. Please move the citation so that it appears before the period, in accordance with standard scientific writing conventions.

Response 6: Thank you very much for your careful review. We have made the necessary corrections to the illustrations and table references.

Comments 7: Why is the aim of the study repeated in the Methods section? The sentence “The aim of this study was to systematically summarize the chemical constituents of cranberry and their extraction methods” (lines 150–152) is already stated at the end of the Introduction. Please rephrase or remove this sentence to avoid unnecessary repetition.

Response 7: Thank you very much for your suggestions. We changed “The aim of this study was to systematically summarize the chemical constituents of cranberry and their extraction methods” to “The article summarizes extraction methods for different categories of chemical constituents in cranberries”. We have removed these duplicate statements.

Comments 8: A section titled “3. Results” should be added, and the “Search results” subsection should be placed under this main Results section. All subsequent subsections, such as Cranberry Composition and Cranberry Pharmacology, should then be numbered accordingly under Results. You should not start directly with “3. Cranberry chemical composition” without first presenting the main Results heading.

Response 8: Thank you very much for your suggestions. We have incorporated the article into “3. Results” as requested and revised the numbering of all section headings. We hope this meets your requirements.

Comments 9: The authors state that they prioritized citing only two publications in Table 1 (references 17 and 19) because “these two publications were more comprehensive and thus more valuable for reference.” If this approach is maintained, the authors should explicitly mention this choice in the text, explaining the rationale for including only these two references. Alternatively, they should include the other relevant references in the table.

Response 9: Thank you very much for your suggestion. We have added references to References 17 and 19 on line 205 of the article, clarifying the sources of the chemical composition data for different cranberry varieties in Table 1. Additionally, on line 246, we have provided clarification regarding the citation of data in Table 2.

Dear Editors and Reviewers,

We extend our sincerest wishes for the utmost success in your pursuits. We are profoundly grateful for your meticulous examination of our manuscript amidst your demanding schedule, and for the esteemed recommendations you have proffered. Each of your comments holds significant value and has been instrumental in refining and enhancing our work, as well as serving as a critical guidepost for our research endeavors. Subsequently, we have furnished comprehensive replies to the critiques offered by the reviewers. We have endeavored to ensure that all modifications are transparently presented, and we trust that the revised version of our manuscript now fulfills the criteria for publication.

Sincerely,

Jiao Xu

Jiamusi College, Heilongjiang University of Traditional Chinese Medicine,

Jiamusi, China

Email: xujiao2023@163.com